# Differences in size and number of embryonic type II neuroblast lineages correlate with divergent timing of central complex development between beetle and fly

**Simon Rethemeier[1,2], Sonja Fritzsche[1], Dominik Mühlen[1], Gregor Bucher[1], Vera S Hunnekuhl[1]***

[1]University of Göttingen, Johann-Friedrich-Blumenbach Institute, GZMB, Department of Evolutionary Developmental Genetics, Göttingen, Germany; [2]University Medical Center Göttingen (UMG), Göttingen, Germany

## eLife Assessment

The study is a **valuable** contribution to the question of evolutionary shifts in neuronal proliferation patterns and the timing of developmental progressions. The authors present **convincing** data which confirm the presence of type II NB lineages in beetle with the same molecular characteristics as the *Drosophila* counterparts but differing in lineage size and number. The data lay the foundation for future analysis of the role and molecular characteristics of individual lineages and of whether differences in the identity, proliferation pattern, and timing of developmental progression can be linked to differences in the development of functionality of the central complex.

***For correspondence:**
vera.terblanche@uni-goettingen.de

**Competing interest:** The authors declare that no competing interests exist.

**Abstract** The insect brain and the timing of its development underwent evolutionary adaptations. However, little is known about the underlying developmental processes. The central complex of the brain is an excellent model to understand neural development and divergence. It is produced in large parts by type II neuroblasts, which produce intermediate progenitors, another type of cycling precursor, to increase their neural progeny. Type II neuroblasts lineages are believed to be conserved among insects, but little is known on their molecular characteristics in insects other than flies. *Tribolium castaneum* has emerged as a model for brain development and evolution. However, type II neuroblasts have so far not been studied in this beetle. We created a fluorescent enhancer trap marking expression of *Tc-fez/earmuff*, a key marker for intermediate progenitors. Using combinatorial labeling of further markers, including *Tc-pointed*, we characterized embryonic type II neuroblast lineages. Intriguingly, we found nine lineages per hemisphere in the *Tribolium* embryo while *Drosophila* produces only eight per brain hemisphere. These embryonic lineages are significantly larger in *Tribolium* than they are in *Drosophila* and contain more intermediate progenitors. Finally, we mapped these lineages to the domains of head patterning genes. Notably, *Tc-otd* is absent from all type II neuroblasts and intermediate progenitors, whereas *Tc-six3* marks an anterior subset of the type II lineages. *Tc-six4* specifically marks the territory where anterior-medial type II neuroblasts differentiate. In conclusion, we identified a conserved pattern of gene expression in holometabolan central complex forming type II neuroblast lineages, and conserved head patterning genes emerged as new candidates for conferring spatial identity to individual lineages. The higher number and greater lineage size of the embryonic type II neuroblasts in the beetle correlate with a previously

described embryonic phase of central complex formation. These findings stipulate further research on the link between stem cell activity and temporal and structural differences in central complex development.

## Introduction

Neural development of insects allows to study molecular and cellular principles in easy to manipulate invertebrate models. The fruit fly *Drosophila melanogaster* (*Drosophila* hereafter) has mostly been used for this purpose, making it one of the best understood models for neurogenesis (*Konstantinides et al., 2018*; *Li et al., 2014*). However, it has remained a puzzle how the huge ecological diversity of insects and the divergent neural anatomies that are adapted to different niches (*Farnworth et al., 2020b*; *Couto et al., 2020*) evolved. Furthermore, *Drosophila* may in many instances not be a good representation of insect development and some processes are derived in the fly lineage (*Hakeemi et al., 2022*; *Klingler and Bucher, 2022*; *Tautz et al., 1994*). For these reasons, the beetle *Tribolium castaneum* (*Tribolium* hereafter) has been introduced as an additional model for insect neural develop-ment (*Biffar and Stollewerk, 2014*; *Biffar and Stollewerk, 2015*; *Koniszewski et al., 2016*; *Posnien et al., 2023*). *Tribolium* is a grain storage pest and all stages live in flour; hence, there is no major transition of habitat or lifestyle between the larval and adult stage as in flies. However, compared to *Drosophila,* the larvae are more mobile through the use of walking legs, which allows them to navigate within or between food sources (*Farnworth et al., 2020b*; *Klingler and Bucher, 2022*; *Koniszewski et al., 2016*). In this beetle, many molecular genetic manipulation and labeling methods have been established (*Gilles et al., 2015*; *Gilles et al., 2019*; *Posnien et al., 2009*; *Farnworth et al., 2020a*; *Hunnekuhl et al., 2020*). Specifically, labeling of neural cell types in *Tribolium* has been facilitated by the advent of CRISPR-Cas9 (*Gilles et al., 2015*; *Farnworth et al., 2020a*; *He et al., 2019*). Another factor that makes *Tribolium* an informative model for the understanding of insect brain develop-ment is the more conserved development of the head neuroectoderm facilitating the identification of

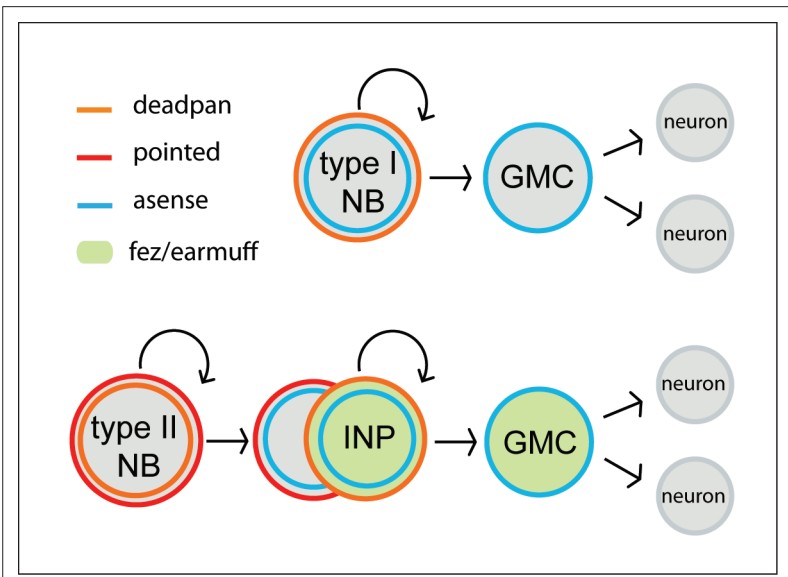

**Figure 1.** Neuroblasts (NBs) of the *Drosophila* nervous system. Type I NBs (top panel) and type II NBs (lower panel). Type I NBs undergo stem cell-like divisions, with each round producing a ganglion mother cell (GMC) that divides one more time to produce two neurons or glia cells. In *Drosophila*, all type I NBs express *dpn* and *ase*, whereas GMCs express only *ase*. Type II NBs express *dpn* and *pnt* but not *ase*. They also undergo repeated divisions, with each round producing an intermediate progenitor cell (INP), which is also a proliferating progenitor and expresses *ase*. INPs undergo a maturation process during which *pnt* is still expressed initially, but not *dpn*. Mature INPs express *dpn*, *ase,* and *fez/erm*, whereas GMCs express *ase* and *fez/erm*. Type II NB lineages are only found in the anterior brain; they produce central complex neurons and glia (*Bayraktar and Doe, 2013*; *Southall and Brand, 2009*).

specific neurogenic domains and allowing cross-species comparisons (*Posnien et al., 2023*; *Posnien et al., 2010*; *Posnien et al., 2011b*).

The insect central complex is an anterior, midline spanning neuropile and constitutes an important brain center for the processing of sensory input, coordination of movement, and navigation (*Honkanen et al., 2019*; *Pfeiffer and Homberg, 2014*). The highly conserved basic structure of the central complex, along with diverse specifications in response to ecological requirements among different insect species, makes this brain structure a very interesting model for evolutionary developmental studies (*Farnworth et al., 2020b*; *Koniszewski et al., 2016*; *Honkanen et al., 2019*). Between the two insect models *Drosophila* and *Tribolium,* a temporal shift in the emergence of central complex neuropile was observed, with *Tribolium* developing a functional larval central complex during embryogenesis, whereas it develops only at the onset of adult life in *Drosophila* (*Farnworth et al., 2020b*; *Koniszewski et al., 2016*). It is, however, not known how these temporal differences are established during development on a cellular and molecular level.

The entire insect central nervous system including the brain is produced by neural stem cells, the neuroblasts (NBs) (*Hartenstein and Stollewerk, 2015*). While there are evolutionary modifications in the relative position of trunk NBs and their gene expression profiles between fly and beetle, the core determinants specifying their role as neural progenitors are conserved (*Biffar and Stollewerk, 2014*). NBs undergo repeated divisions producing rows of ganglion mother cells (GMCs), which divide one more time to produce neurons and/or glia (see *Figure 1*, top panel). This leads to neural cell lineages that include the NB itself and its progeny (*Doe and Skeath, 1996*; *Stollewerk, 2016*). Several NBs of the anterior-most part of the neuroectoderm contribute to the central complex, which has been characterized as the most complex brain structure (apart from the optic lobe) (*Boyan and Reichert, 2011*; *Strausfeld, 2012*). Intriguingly, in *Drosophila* and in the grasshopper *Schistocerca gregaria,* an NB subtype, the type II NBs, were found to prominently contribute to the formation of the central complex (*Boyan and Reichert, 2011*; *Boyan et al., 2017*). These specific NBs generate more offspring by producing another class of neural precursors, the intermediate progenitors (INPs), which also divide in a stem cell-like fashion (*Walsh and Doe, 2017*; *Bayraktar and Doe, 2013*; *Figure 1*, lower panel). Type II NBs and INPs have attracted a lot of attention because intermediate, cycling progenitors have also been described in vertebrates, the radial glia cells, which produce a variety of cell types (*Malatesta et al., 2008*; *Barry et al., 2014*). In addition, in *Drosophila,* brain tumors have been induced from type II NB lineages (*Caussinus and Gonzalez, 2005*), opening up the possibility of modeling tumorigenesis in an invertebrate brain, thus making these lineages one of the most intriguing stem cell models in invertebrates (*Homem and Knoblich, 2012*; *Bowman et al., 2008*).

In the developing insect brain, the number of neurons that is produced by each type II NB is significantly larger than the offspring of a type I NB. Medial *Drosophila* type II NBs produce an average of 450 neurons, whereas type I NBs of the brain only generate 100–200 neurons each (*Bello et al., 2008*; *Boyan and Williams, 2011*; *Izergina et al., 2009*). This increased number of neural cells that are produced by individual type II lineages, along with the fact that NB lineages can produce different types of neurons, leads to the generation of extensive neural complexity within the anterior insect brain (; *Bayraktar and Doe, 2013*; *Southall and Brand, 2009*).

Based on descriptions of large proliferative lineages in the grasshopper *S. gregaria,* a hemimetabolous insect distantly related to flies, it is widely believed that type II NBs and INPs are conserved within insects (*Boyan and Reichert, 2011*; *Boyan et al., 2010*). However, a molecular characterization of such lineages that, on the one hand, will allow to test whether the cell type markers characterized in *Drosophila* are more widely conserved, and, on the other hand, will facilitate an identification of subcell types, has not been performed in an insect other than *Drosophila*.

Whereas it is widely believed that *Tribolium* does use type II NB in central complex formation (*Farnworth et al., 2020b*; *He et al., 2019*; *Garcia-Perez et al., 2021*), their presence has so far not been unequivocally shown and a thorough comparison of type II NB lineages and their subcell types between the fly and the beetle model has been lacking.

The characterization of type II NBs in *Drosophila* has in large parts been based on a specific reporter line in which eGFP expression is driven by an *earmuff (fez/erm)* enhancer element and marks INP lineages (*Bayraktar and Doe, 2013*; *Pfeiffer et al., 2008*; *Weng et al., 2010*). *Tribolium earmuff* has been first described as *Tc-fez*, referring to the vertebrate *earmuff* ortholog *fez* (*Posnien et al., 2011b*). We will use *Tc-fez/erm* in the following. A defined sequence of gene expression has been described in

*Drosophila* type II NBs, INPs and GMCs. The ETS-transcription factor *pointed* (*pnt*) marks type II NBs (*Xie et al., 2016*; *Zhu et al., 2011*), which do not express the type I NB marker *asense* (*ase*) but the neural gene deadpan (*dpn*) (*Figure 1*). INPs express *fez/erm* and *ase* and at a more mature stage also *dpn*, whereas *prospero* (*pros*) marks mature INPs and GMCs (*Bayraktar and Doe, 2013*; *Bayraktar et al., 2010*). *Six4* is another marker for *Drosophila* type II NBs and INPs (*Chen et al., 2021*).

In the present work, we characterize type II NBs in *Tribolium* with respect to their number, their location, and the conservation of the molecular markers known from *Drosophila*. We also wanted to test in how far the earlier emergence of the central complex in beetles would be reflected by a change of division activity of type II NBs or INPs. To that end, we used CRISPR-Cas9-induced genome editing for creating a *Tc-fez/erm* enhancer trap line to characterize embryonic *Tc-fez/erm*-expressing cells, including INPs and GMCs of type II NB lineages, and combined this line with other labeling methods, including multicolor *hybridization chain reaction* (HCR) (*Choi et al., 2018*; *Choi et al., 2010*). We found that the *Tribolium* embryo produces nine *Tc-pnt*-expressing type II NBs compared to only eight type II NB lineages in the *Drosophila* embryo (*Álvarez and Díaz-Benjumea, 2018*). We show that these lineages produce central complex cells and confirm a largely conserved molecular code that differentiates INPs of distinct maturation stages and GMCs. Intriguingly, we found that lineage sizes in *Tribolium* embryos are considerably larger and include more mature INPs than in the *Drosophila* embryo (*Álvarez and Díaz-Benjumea, 2018*). We also show that the placodal marker gene *Tc-six4* characterizes the embryonic tissue that gives rise to the larger anterior group of type II NBs, whereas

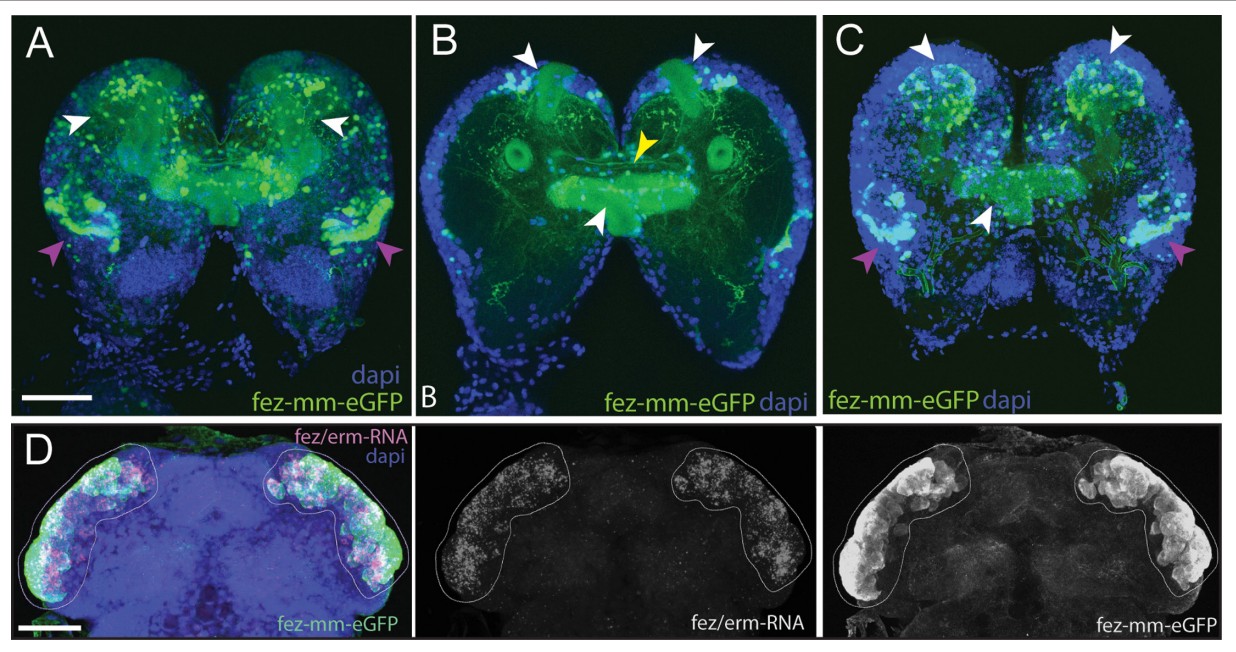

**Figure 2.** Magic mushrooms (fez-mm-eGFP) reporter line. (**A–C**) eGFP-expression in a larval instar brain. (**D**) EGFP antibody labeling in combination with *Tc-fez/erm* RNA in situ in the embryonic head. (**A–D**) DAPI labeling of nuclei. Scale bars in (**A**) and (**D**) are 50 μm. (**A**) Full projection of whole brain scan (fourth-instar larvae): fez-mm-eGPF expression marks many cells of the mushroom bodies (white arrowheads) and of the larval optic lobe (purple arrowheads). Note that we use the terms dorsal and ventral with respect to the neuraxis of the larva. See *Koniszewski et al., 2016* for orientation. (**B**) Central part of the whole brain scan (projection of substack, same brain as in **A**): mm-eGFP marks the peduncles of the mushroom bodies (white arrowheads) and some central complex ensheathing cells (probably glia cells; yellow arrowhead). (**C**) Projection of the dorsal part of same brain as in (**A**) shows calyces and peduncles of mushroom bodies (white arrowheads), and optic neuropile (purple arrowheads). (**D**) Extensive co-localization of fez-mm-eGFP with *Tc-fez/erm*-RNA in white encircled area, which gives rise to protocerebral structures. Projection of all planes where expression was detected, embryonic stage NS11. *Figure 2—figure supplement 1* and *Supplementary file 1a and b* give details of the generation and sequence-based characterization of this line.

The online version of this article includes the following source data and figure supplement(s) for figure 2:

**Figure supplement 1.** *Tc-fez/erm* locus with inserted reporter construct.

**Figure supplement 1—source data 1.** Uncropped image of agarose DNA-Gel including the part shown in *Figure 2—figure supplement 1B* (left).

**Figure supplement 1—source data 2.** Uncropped image of agarose DNA-Gel including the part shown in *Figure 2—figure supplement 1B* (right).

*Tc-six3* is marking only an anterior subset of these lineages. Interestingly, there is a part of the central complex precursors expressing neither *Tc-six3* nor *Tc-otd,* which is absent from all type II NB lineages.

## Results

### A CRISPR-Cas9-NHEJ-generated enhancer trap lines marks *Tc-fez/erm-*expressing cells at the embryonic and larval stages

We created an enhancer trap line driving eGFP that reflects *Tc-fez/erm* gene expression in the embryo. We named the line *fez magic mushrooms* (fez-mm-eGFP) because it marks the mushroom bodies of the larval brain (*Figure 2A–C*). The reporter construct was inserted 160 bp upstream of the *fez/erm*-transcription start site using the *non-homologous end joining* (NHEJ) repair mechanism (see *Figure 2—figure supplement 1A* for scheme). Enhancer traps sometimes only reflect parts of a gene expression pattern; therefore, we carefully evaluated eGFP co-expression with *Tc-fez/erm*-RNA. We found that at the embryonic stage fez-mm-eGFP co-localized with most *Tc-fez/erm*-expressing cells in an area that will give rise to the protocerebral structures mushroom bodies, larval eyes, and central complex (*Posnien et al., 2023*; *Posnien et al., 2011b*; *Figure 2D*). The fez-mm-eGFP line marked several progenitor cell types, including cells that we characterize as type II NB-derived INPs (see below). Based on these analyses, we use the fez-mm-eGFP expression as a reporter for *Tc-fez/erm* expression.

### *Tc-pointed* marks a population of nine type II NBs that are associated with fez-mm-eGFP marked INP lineages

To identify type II NBs, we searched for cell groups that express the markers known from flies. The transcription factor *pointed* (*pnt*) marks type II NBs in *Drosophila* and is required for their correct differentiation (*Xie et al., 2016*; *Zhu et al., 2011*). In both *Tribolium* and *Drosophila*, *pnt* is also expressed in other areas (this work/*Brunner et al., 1994*; *Klämbt, 1993*). Therefore, we looked for *Tc-pnt* expression in large cells with large nuclei (typical NB morphology) that were closely associated with *Tc-fez/erm*-expressing cells (marking putative INPs). We found a total of nine such clusters instead of the eight expected from flies (*Figure 3A–C*). We also found that the large *Tc-pnt*-expressing cells as well as fez-mm-eGFP-expressing cells are mitotically active (*Figure 3C–I*). Further molecular and cell size analysis corroborated our interpretation that these cells constitute type II NBs, INPs at different maturation stages, and GMCs (see details below).

Looking at a series of embryonic stages, we determined the emergence and further development of *Tc-pnt* and fez-mm-eGFP-expressing lineages as well as their arrangement with respect to the embryonic head and to one another. We first detected conspicuous adjacent expression of *Tc-pnt* and fez-mm-eGFP at stage NS7 (not shown). At stage NS11, seven *Tc-pnt*-clusters are present in a horse-shoe-shaped formation surrounding fez-mm-eGFP-positive cells (white arrows in *Figure 3A*, star marks one cluster out of focus). At stage NS13, these groups are found in a position at the medial border of the head lobes and their number has increased to nine. The seven clusters that were already present at stage NS11 are arranged in an anterior group (white arrowheads in *Figure 3B*) at stage NS13 while two additional clusters have emerged more posteriorly (yellow arrows in *Figure 3B–I*). The clusters are in different depths (reflected by the three optical sections shown in *Figure 3B–I to B-III*, clusters out of focus are visualized by gray circles in *Figure 3B'I–II*, respectively). Fez-mm-eGFP expression not related to type II NB offspring was found in the lateral head lobe from NS13 onward (blue arrowheads in *Figure 3*). At stage NS14, six of the anterior-medial clusters are aligned in one plane along the medial rim of the developing brain (*Figure 3C–I*), whereas the most posterior cluster of the anterior group is located at a separate position and in a deeper plane (*Figure 3C-II*). The posterior group is found at about the same focal plane as well (*Figure 3C-II*). The aligned six clusters produce their *Tc-fez/erm*-positive offspring toward laterally, whereas the *Tc-fez/erm*-positive offspring of the more posterior cluster are found medially of it (*Figure 3C-II*). The two most posterior clusters produce *Tc-fez/erm*-cells in an anterior-lateral direction (*Figure 3C-II*). The apparent rearrangement of the *Tc-pnt*-clusters from the anterior rim of the embryonic head to a medial position reflects not active migration but follows the overall tissue movements during head morphogenesis that was previously described in *Tribolium* (*Posnien et al., 2011b*; *Figure 3D*). Based on their position by the end of embryogenesis (stage NS14), we assign the nine type II NB clusters into two groups: one anterior

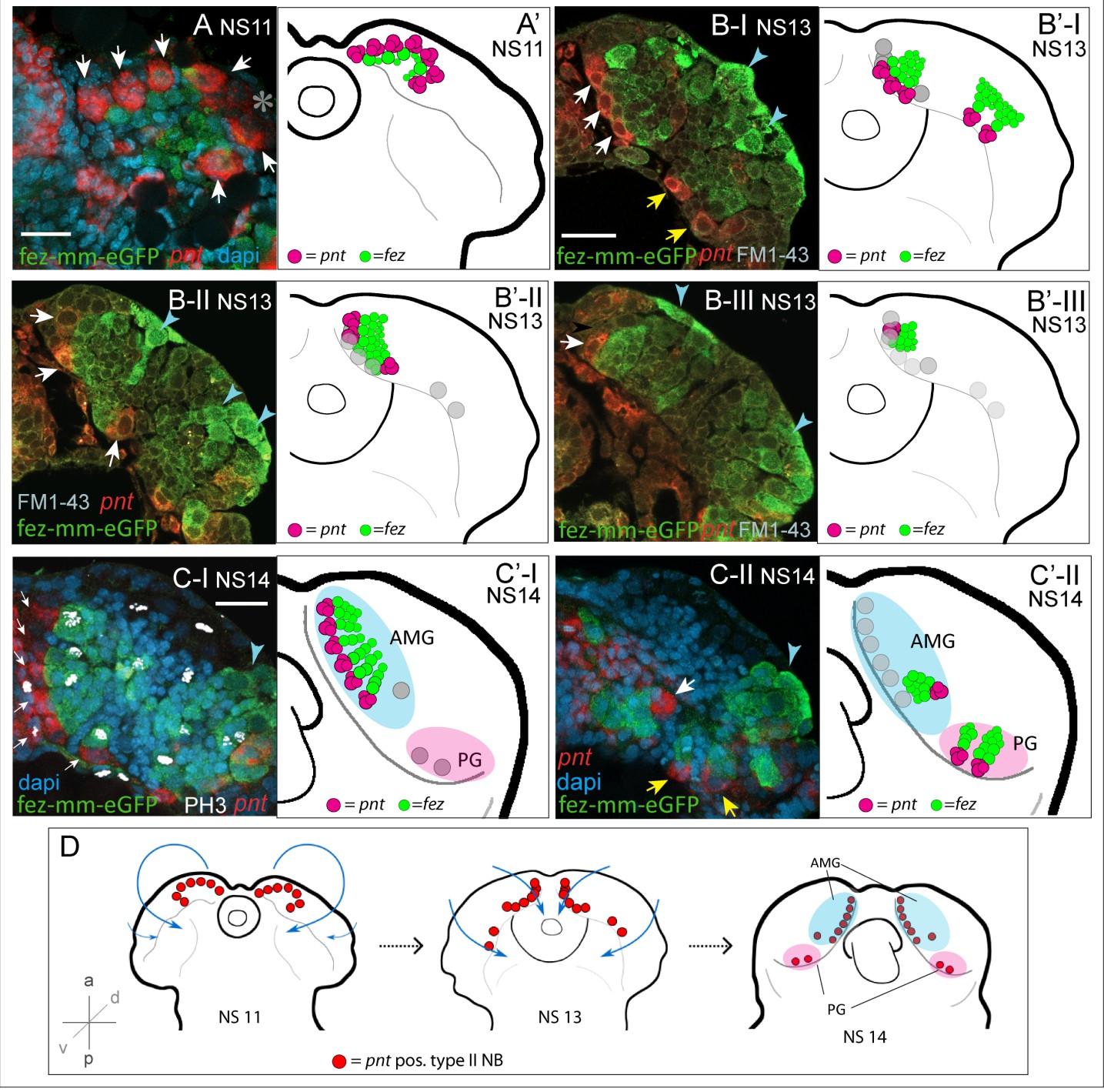

**Figure 3.** Developmental occurrence and progression of *Tc-pnt*-clusters and associated fez-mm-eGFP cells (type II neuroblast [NB] lineages). (**A–C**) Anti-GFP antibody staining reflecting fez-mm-eGFP expression in combination with *Tc-pnt* RNA in situ hybridization and DAPI labeling of nuclei (**A, C**) or FM1-43 membrane staining (**B**), single planes from confocal z-stacks of right head lobes. Scalebars are 25 μm. (**A'–C'**) Schematic drawing of the head lobe at the respective stages; clusters out of focus depicted by gray circles. (**D**) Schematic depiction of the localization of type II NBs during *Tribolium* head morphogenesis. (**A**) At stage NS11, seven *Tc-pnt*-clusters (white arrows) are found in an anterior-medial horseshoe-like arrangement with fez-mm-eGFP cells in the center. Asterisk indicates the position of one *Tc-pnt*-cluster, which is outside the focal plane of this image. (**B**) At stage NS13, the seven anterior-medial clusters have been moved to the medial margin of the developing brain by morphogenetic movements of the head lobes (white arrows). The clusters are situated in different dorsoventral levels shown in the different images (**B-I**) to (**B-III**). Two additional posterior *Tc-pnt*-clusters have appeared (yellow arrows). Blue arrows point to fez-mm-eGFP-positive cells that are not part of the type II NB lineages. (**B-I**) Ventral level; (**B-II**) mid-level; (**B-III**) dorsal level. (**C**) At stage NS14, a group of six clusters are arranged in one plane anterior-medially in the brain lobe (white arrows) (**C-I**). Anti-PH3

*Figure 3 continued on next page*

Figure 3 continued

labeling marks mitotic cells within the lineages (white signal) and clusters out of the focal plane are indicated by gray circles in (**C'-I**). (**C-II**) One cluster of the anterior median group (white arrow) is in a deeper plane, as well as the two posterior clusters (yellow arrows). Blue arrowheads point to lateral fez-mm-eGFP expression not associated with type II NBs. (**A'-C'**) AMG = anterior-medial group (blue), PG = posterior group (pink). Gray circles in (**B'-C'**) indicate position of type II NBs in different focal planes. (**D**) *Tc-pnt*-expressing cells identified as type II NBs first appear bilaterally in the anterior-most part of the embryonic head (stage NS11). During head morphogenesis, this anterior tissue folds over so that the type II NBs end up in a more medial position. Two more type II NBs emerge posteriorly (stage 13). At the end of embryogenesis (stage NS14), the type II NBs can be divided in an AMG (blue; including type II NBs 1–7, numbering starting from anterior) and a PG (pink, including type II NBs 8–9).

median group consisting of seven type II NBs and one posterior group consisting of two type II NBs (see *Figure 3C and D*).

## Characterization of cell and nuclear size of *pnt*-positive type II NBs

We wanted to confirm that each of the *Tc-pnt* and *Tc-fez/erm*-expressing lineages contains an NB. NBs, including type II NBs, are larger than other cells, have larger nuclei, and are mitotically active (*Boyan et al., 2010*). Therefore, we determined the cell size of the largest cells of the *Tc-pnt* cluster and compared it to a random sample of cells of the embryonic head. Both samples were taken from stage NS13 and NS14 embryos. We found that the cells that we had assigned as type II NBs had a significantly larger diameter (av.: 12.98 μm; n=43) than cells of the reference sample (av.: 6.75 μm; n=1579) (*Figure 4*). We also found that the diameters of nuclei of type II NBs were significantly larger (av.: 8.73 μm; n=17) than the ones of a control sample (av.: 5.21 μm; n=1080) (*Figure 4*). Although type II NBs were clearly larger than the average cell of the head lobes, the control sample also contained cells of an equal size (*Figure 4*), which may constitute type I NBs.

## Conserved patterns of gene expression mark *Tribolium* type II NBs, different stages of INPs and GMCs

*Drosophila* type II NBs, INPs, and GMCs express a specific sequence of genes, including *asense* (*ase*), *deadpan* (*dpn*), and *prospero* (*pros*) (*Figure 1*; *Bayraktar and Doe, 2013*). However, these markers have up to now not been tested for expression in type II lineages in other organisms. We tested if the respective lineages express these markers in a comparable sequence in the beetle and if they mark distinctive cell types within the lineages (*Figures 5A–D and 6A and B*). We found that the large *Tc-pnt*-positive but fez-mm-eGFP-negative cells (i.e., type II NBs) express the gene *Tc-dpn*, in line with expression of this gene in *Drosophila* neural precursors (*Figure 5A*). Like *Drosophila* type II NBs, these cells do not express the type I NB marker *Tc-ase* (*Figure 5D*, yellow arrowhead). We further found additional cells directly adjacent of the type II NBs itself, which we believe are recently born immature INPs. They are *Tc-pnt*-positive but fez-mm-eGFP-negative and neither express *Tc-ase* (*Figure 5D.*, pink arrowheads). Expression of *Tc-ase* starts in the following in immature *Tc-pnt*-positive INPs (*Figure 5D*, blue arrowhead) and is maintained in INPs that are *Tc-pnt*-negative but are marked by fez-mm-eGFP (*Figure 5D and E*). *Tc-dpn* is absent from immature *Tc-pnt*-positive INPs (*Figure 5D*, pink arrowhead) but is expressed in mature INPs (*Figure 5B*, orange arrowheads). Cells that are located at the distal end of the lineages that do not express *Tc-dpn* but are positive for fez-mm-eGFP and *Tc-ase* are classified as GMCs (*Figure 5B, C and E*). We found the gene *Tc-pros* expressed in most fez-mm-eGFP expressing cells of a lineage (in *Figure 6A and B*) and infer from the extent of the expression that it is expressed in mature INPs (*Tc-dpn+*) and GMCs (*Tc-dpn-*). Fez-mm-eGFP-positive cells at the base of the lineage that do not express *Tc-pros* are immature INPs (*Figure 6A and B*, blue arrowheads). In summary, the expression dynamics found in the *Tribolium* type II NB lineages are very similar to the one found in *Drosophila*. Therefore, these neural markers can be used for a classification of cell types within the lineages into type II NBs (*Tc-pnt+*, *Tc-fez/erm-*, *Tc-ase-*, *Tc-dpn+*, *Tc-pros-*), immature-I INPs (*Tc-pnt+*, *Tc-fez/erm-*, *Tc-ase-*, *Tc-dpn-*, *Tc-pros-*), immature-II INPs (*Tc-pnt+*, *Tc-fez/erm+*, *Tc-ase+*, *Tc-dpn-*, *Tc-pros-*), mature INPs (*Tc-pnt-*, *Tc-fez/erm+*, *Tc-ase+*, *Tc-dpn+*, *Tc-pros+*), and GMCs (*Tc-pnt-*, *Tc-fez/erm+*, *Tc-ase+*, *Tc-dpn-*, *Tc-pros+*). This classification is summarized in *Figure 7A and B*. Image stacks from specimens co-stained with the anti-PH3 mitosis marker (data shown in *Figure 3C* and listed in *Tables 1 and 2*, available at https://doi.org/10.25625/8IVICL) indicate that immature-I and -II INPs are not dividing, whereas mature INPs are.

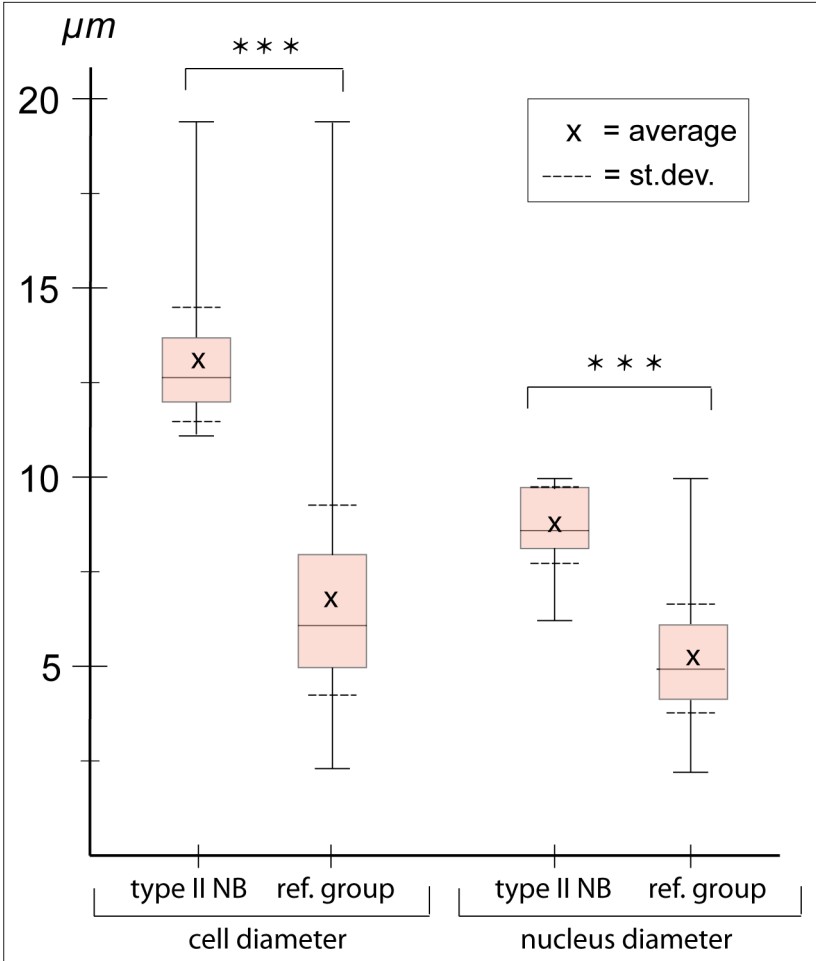

**Figure 4.** Cell and nuclear diameter of type II neuroblasts (NBs) compared to a control group. Stages NS13-14. Left: diameter of *Tc-pnt*-pos. type II NBs (average diameter = 12.98 µm; n=43) compared to the control group (average diameter = 6.75, n=1579). Right: average nuclear diameter of type II NBs (=8,73 µm, n=17) compared to the average nuclear diameter of the control group (=5.21, n=1080). Differences between groups are significant (*t*-test; *** ≙p<0.001).

## Type II NB lineages produce central complex cells that are marked by shaking hands (skh)

To test if the *Tribolium* embryonic type II NB lineages contribute to the beetle central complex like in the fly, we used the shaking hands- (skh-) reporter line that marks central complex neurons and their postmitotic embryonic precursors (*Garcia-Perez et al., 2021*). By co-staining for *Tc-pnt* and *Tc-fez/erm* RNA in the background of the skh-line, we found that the skh-positive cells are located directly distally of the type II NB lineages in a position where the type II NBs-derived neural cells are expected (*Figure 8A, B, E and F*). Most skh-cells have switched off *fez/erm*-expression, but at the transition between *Tc-fez/erm* and skh cells we found some cells that express both markers (*Figure 8D and E*), supporting the view that many skh-positive central complex cells stem from the *fez*-expressing GMCs of the type II NB lineages (*Figure 8E*). We found skh cells at the end of lineages of the anterior-median group (*Figure 8A*) and at the end the two posterior lineages (*Figure 8B*). We therefore conclude that both groups contribute to central complex neuropile (*Figure 8F*). We cannot say whether the lineage of type II NB seven, which we assigned to the anterior-median group, but which is located with some distance to the other type II NB of that group (*Figure 3C and D*; gray circle in *Figure 8F*), produces skh+ cells, but we assume that it does.

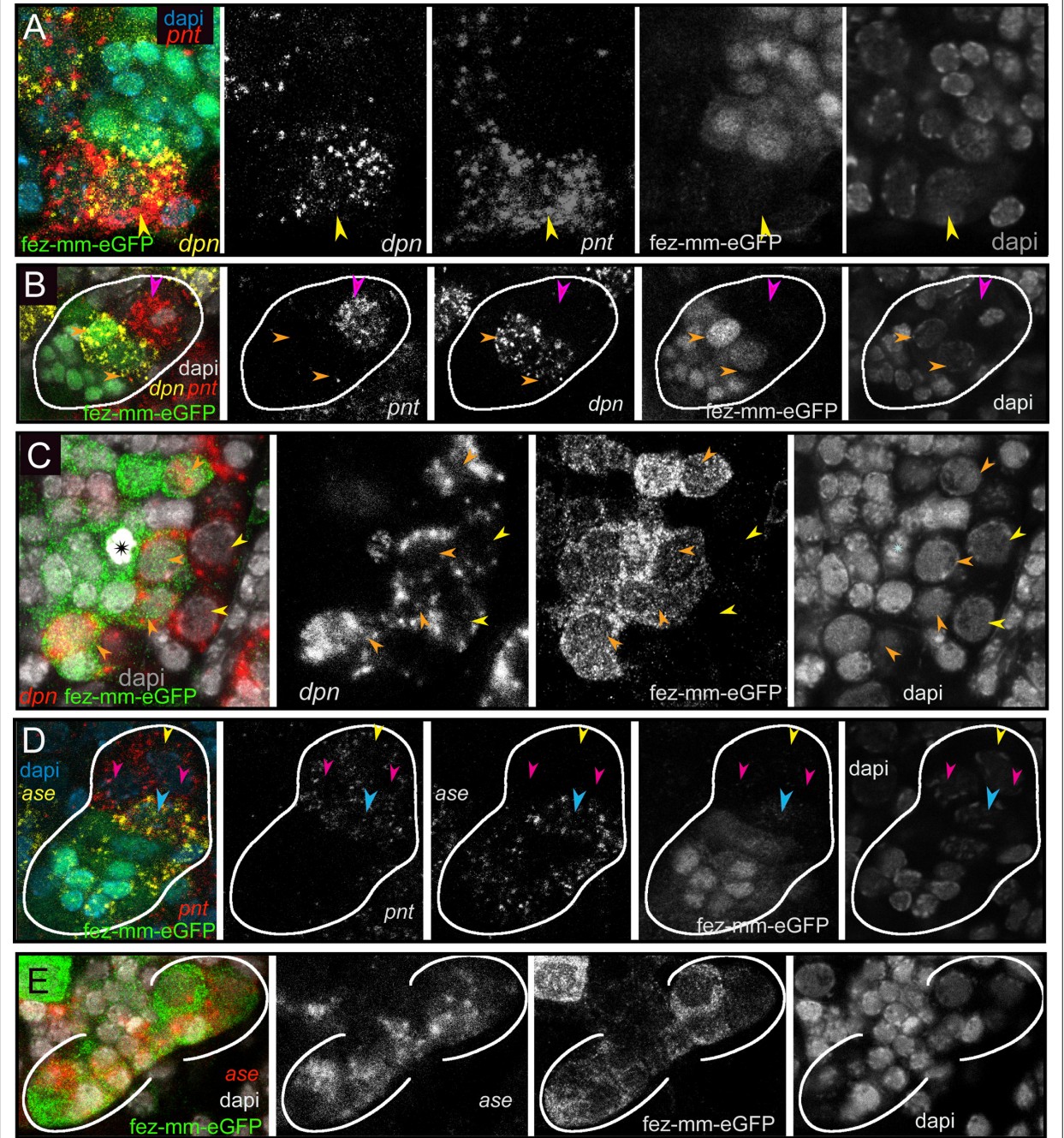

**Figure 5.** Differential expression of the neural markers *Tc-pnt, Tc-fez/erm, Tc-dpn,* and *Tc-ase* in type II neuroblasts (NBs), intermediate progenitors (INPs), and ganglion mother cells (GMCs). (**A, B, D**) Anti-GFP staining visualizing fez-mm-eGFP expression in combination with HCR-labeling of further factors. (**C, E**) GFP antibody staining in combination with RNA in situ hybridization. (**A–E**) DAPI staining of nuclei. All panels are single planes from confocal z-stacks. (**A**) Stage NS12, expression of *Tc-pnt* and *Tc-dpn* in type II NBs (yellow arrowhead) with adjacent fez-mm-eGFP-positive INPs. (**B**) Stage NS13, white line marks one lineage. Expression of *Tc-dpn* is absent in the immature-I *Tc-pnt* + INPs (pink arrowhead) but present in mature INPs (orange arrowheads), white line marks one lineage. Type II NB not in focus of this image. (**C**) Stage NS14, expression of *Tc-dpn* in type II NBs (yellow arrowheads), and mature INPs (orange arrowheads). Asterisk marks mitosis in a fez-mm-eGFP cell visible through the condensation of chromatin stained by DAPI. (**D**) Stage NS13, *Tc-ase-* and *Tc-pnt*-expression in type II NBs and INPs, white line marks one lineage. Yellow arrowhead: type II NBs (*Tc-pnt+*), pink arrowheads: immature-I INP (*Tc-pnt+*), blue arrowhead: immature-II INP (*Tc-fez+/ Tc-ase+/ Tc-pnt+*). (**E**) Stage NS13, expression of *Tc-ase* in fez-mm-eGFP-positive cells, location of type II NBs and immature-I INPs is outside the focus.

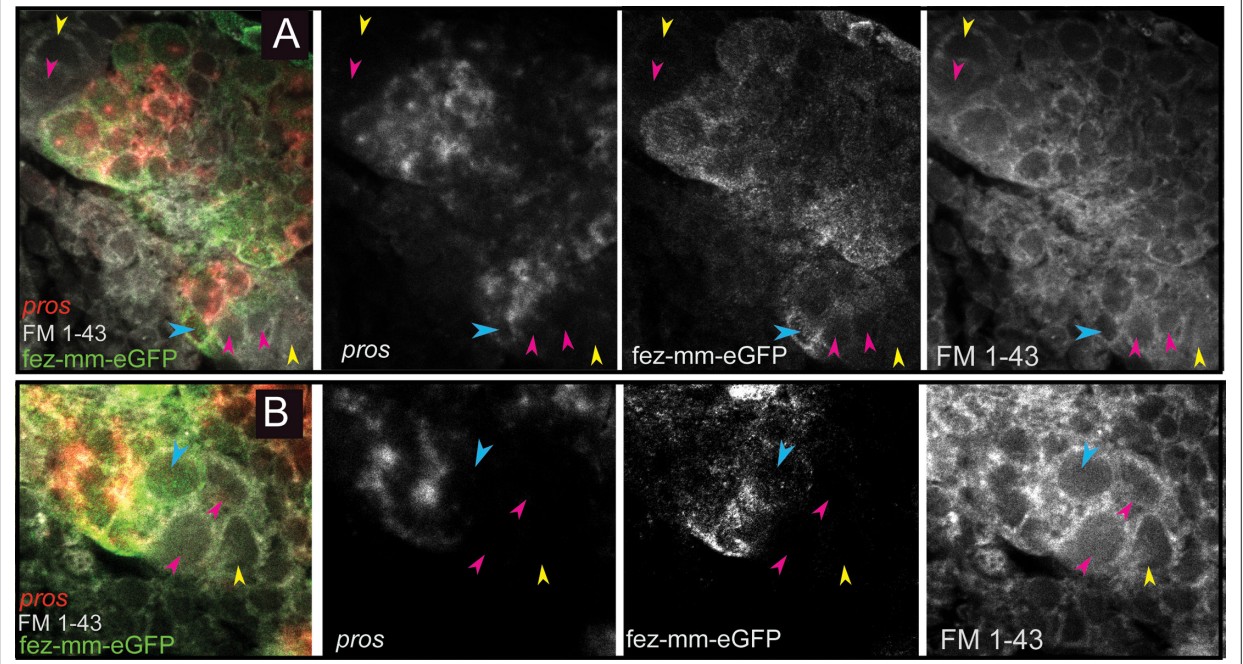

**Figure 6.** Expression of *Tc-pros* in fez-mm-eGFP cells. (**A, B**) Anti-GFP antibody staining in combination with *Tc-pros* RNA in situ hybridization. Membranes stained with FM 1–43. Single planes from confocal z-stacks. (**A**) Stage NS13, two type II NB lineages producing INPs and GMCs in opposing directions. Expression of *Tc-pros* in mature INPs and GMCs. Cluster of type II NBs (yellow arrowheads) and immature-I INPs (pink arrowheads). Immature-II INPs express *Tc-fez/erm* but not *Tc-pros* (blue arrowhead). (**B**) Stage NS13, detail of *Tc-pros*-negative type II NBs (yellow arrowhead) and immature-I INPs (pink arrowheads) and immature-II INP (blue arrowhead). Note that distinction between type II NB and immature INPs in (**A**) and (**B**) is only made based on position.

## The *Tribolium* embryonic lineages of type II NBs are larger and contain more mature INPs than those of *Drosophila*

In beetles, a single-unit central body (most likely constituting the fan-shaped body; *Koniszewski et al., 2016*) develops during embryogenesis and has gained functionality at the onset of larval life (*Farnworth et al., 2020a*). By contrast, in *Drosophila* central complex commissural tracts originating from type II NBs form in the embryo but the type II NB become quiescent and only resume divisions in the late larval stage (*Walsh and Doe, 2017*; *Álvarez and Díaz-Benjumea, 2018*). In stark contrast to *Tribolium,* a functional central complex is only observed in the adult fly (*Farnworth et al., 2020b*). In addition, the volume of the L1 central brain including the central complex is about four times larger in *Tribolium* than it is in *Drosophila* (*Bullinger, 2024*), while there are only small differences in the relative neuropilar volumes of the adult beetle and fly central bodies (*Bullinger, 2024*; *Dreyer et al., 2010*). As type II NBs contribute to central complex development, we asked in how far the embryonic division pattern of these lineages in the beetle would reflect this heterochronic development. To assess the size of the embryonic type II NB lineages in beetles, we counted the *Tc-fez/erm*-positive (fez-mm-eGFP) cells (INPs and GMCs) of the anterior-medial group (type II NB lineages 1–7). As we were not always able to distinguish between cells belonging to neighboring INP-lineages, we quantified and averaged the cell number of all lineages of the anterior-medial group. We also evaluated the proportion of dividing cells in that group based on anti-PH3 staining (*Table 1*).

We found that at stage NS13 (early retracting embryo) each lineage consisted on average of 18.4 progenitor cells and at stage NS14 (retracted embryo) of 16.8 progenitors. Between 5 and 10% of the cells were marked by the PH3 antibody. As PH3 marks dividing cells in a specific phase, the portion of dividing cells may be higher. We further quantified the number of mature *Tc-dpn*-positive INPs (*Table 2*).

We used the numbers from *Tribolium* (*Tables 1 and 2*) for comparison with type II NB lineages from *Drosophila* embryos (data from *Walsh and Doe, 2017*). At fly embryonic stage 13, which is comparable to *Tribolium* stage NS14 in that is represents a retracted germband, lineages only contained one

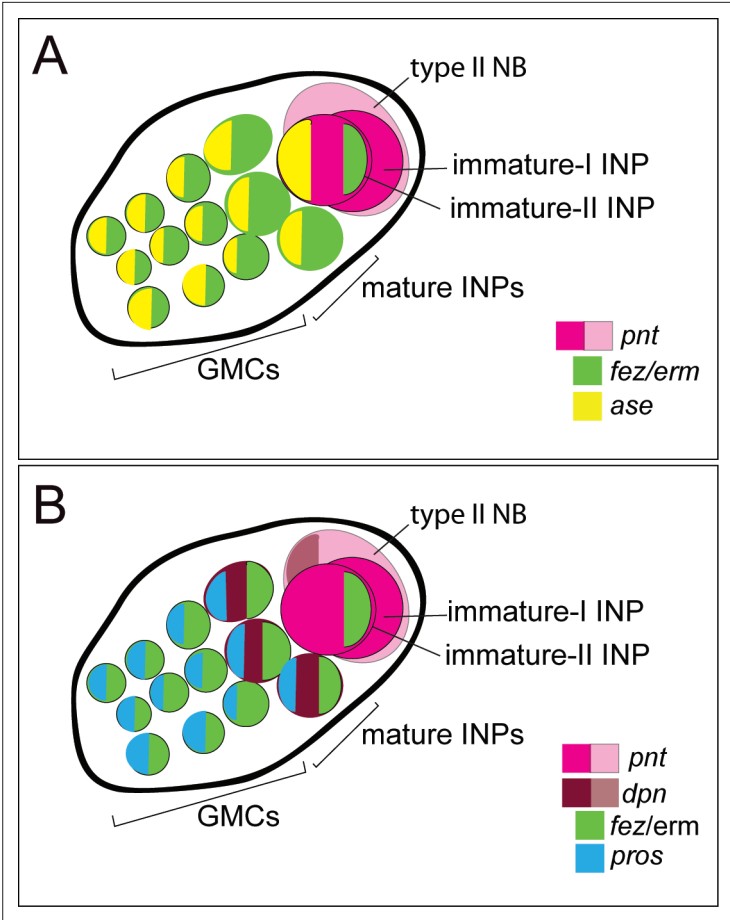

**Figure 7.** Schematic drawing of expression of different markers in a type II neuroblast (NB) lineage. (**A**) and (**B**) show the same lineage with *Tc-fez/erm* and *Tc-pnt* expression but with different additional markers mapped on top. (**A**) Subclassification into type II NBs (*Tc-pnt+*), immature-I INPs (*Tc-pnt+*), immature-II INPs (*Tc-pnt+, Tc-fez/erm+, Tc-ase+*) and mature INPs and GMCs (*Tc-fez/erm+, Tc-ase+*) (based on stainings shown in **Figure 5**). (**B**) Type II NB (*Tc-pnt+, Tc-dpn+*), immature-I (*Tc-pnt+*), and immature-II (Tc-pnt+, *Tc-fez/erm+*). Mature INPs express *Tc-dpn*, *Tc-pros*, and *Tc-fez/erm*, whereas ganglion mother cells (GMCs) express only *Tc-fez/erm* and *Tc-pros* (based on staining shown in **Figures 5 and 6**). Note that of the selected markers immature-I INPs express *Tc-pnt* only.

or more INPs but no GMCs (***Walsh and Doe, 2017***). Therefore, we compared our data for the anterior lineages of *Tribolium* stages NS13 and NS14 to the anterior and median cluster in the *Drosophila* stage 15 and 16 embryos. In both species, these stages are the two penultimate stages of embryogenesis when all type II NB lineages are present, and importantly, in *Drosophila* the maximum lineage sizes and numbers of INPs are reached at these stages. We found that the beetle lineages were significantly larger than the corresponding ones in *Drosophila* (**Figure 9**; *Drosophila* data taken from ***Walsh and Doe, 2017***). Of note, cell counts in *Drosophila* also included neurons (***Walsh and Doe, 2017***). Our counts are based on *Tc-pnt* and fez-mm-eGFP expression and therefore mostly included progenitors. However, based on the small area of co-expression of *Tc-fez/erm* and skh-expression

**Table 1.** *Tc-fez/erm*-positive cell number (type I neuroblasts, intermediate progenitors, and ganglion mother cells).

| Embryo #/stage | Total number of cells of seven anterior lineages | % dividing cells (PH3 antibody) | Cells per lineage (average) |
| --- | --- | --- | --- |
| 1/NS 13 | 146 | 8.2 | 20.9 |
| 2/NS 13 | 111 | 6.3 | 15.9 |
| 3/NS 14 | 139 | 5.0 | 19.9 |
| 4/NS 14 | 96 | 10.5 | 13.7 |

**Table 2.** Mature intermediate progenitors (*Tc-fez/erm* and *Tc-dpn* positive).

| Embryo #/stage | Total number of mature intermediate progenitors of seven anterior lineages | % dividing cells (PH3 antibody) | Number of mature intermediate progenitors per lineage (average) |
|---|---|---|---|
| 1/NS 13 | 24 | n.d. | 3.4 |
| 2/NS 13 | 12 | n.d. | 1.7 |
| 3/NS 13 | 23 | n.d. | 3.3 |
| 4/NS 14 | 27 | 3.7 | 3.9 |
| 5/NS 14 | 30 | 13.3 | 4.3 |

(see above/*Figure 8*) they may also include some newly born neurons, but the counts in *Drosophila* included more mature neurons, which further increases the difference in the number of progenitor cells between the two species and the observed difference in lineage size represents a minimum estimate.

Due to a lower quality of the image data from the posterior group which is in a deeper layer (see *Figure 3*), we were not able to quantify cells of this cluster in *Tribolium*. In *Drosophila*, the posterior lineages are larger (av. 10 cells; *Walsh and Doe, 2017*) than the anterior and median ones, but they are still smaller than the anterior lineages of *Tribolium* at the stages that were investigated.

We also found that the embryonic type II NB lineages in *Tribolium* comprised significantly more mature INPs (*Tc-dpn*+/ *Tc-fez*+) than the corresponding lineages in *Drosophila* (*Figure 9*).

The described observations on the higher numbers of progenitor cells in *Tribolium* compared to *Drosophila* correlate with a previously described earlier (i.e., embryonic) formation of a functional central complex in beetles, which is present in the first larval stage (*Farnworth et al., 2020b*).

### Type II NBs and their lineages are differentially marked by the head patterning transcription factors *Tc-six4* and *Tc-six3* but do not express *Tc-otd*

The anterior-most embryonic insect head is divided into different territories by highly conserved developmental transcription factors that give rise to different parts of the protocerebrum (*Posnien et al., 2023*). Hence, we wondered in which of these molecular territories subsets of type II NB lineages are located, as neuroectoderm regionalization genes are candidates for giving specific spatial identities to individual lineages. We tested co-expression with the previously characterized head patterning transcription factors *Tc-otd*, *Tc-six4*, *Tc-six3*, known to be active in the developing protocerebrum (*Tc-otd*) and in the anterior-medial region of the neuroectoderm, where type II NBs emerge (*Tc-six3*, *Tc-six4*) (*Posnien et al., 2023*; *Posnien et al., 2011b*; *Steinmetz et al., 2010*; *Posnien et al., 2011a*). We found no co-expression of fez-mm-eGFP marked cells of both the anterior-medial and posterior lineages with *Tc-otd*. Rather, *Tc-otd* was expressed in the surrounding embryonic head tissue. *Tc-otd* was also absent from the type II NBs themselves (*Figure 10A and B*). However, *Tc-otd* might be expressed in neural cells derived from these lineages as we can detect expression in the central area of the brain lobes where, for instance, skh cells are located (*Figures 8A and F* and *10A and B*). Interestingly, we found that *Tc-six4* is expressed specifically in that area of the head lobe in which the type II NBs clusters 1–6 of the anterior-medial group first differentiate. At stage NS13, *Tc-six4* marks both, type II NBs and fez-mm-eGFP-expressing INPs and some surrounding cells (*Figure 11A*). At the following differentiation stage (NS14), *Tc-six4* still marks the type II NBs clusters of the anterior-medial group but is not expressed in the fez-mm-eGFP INPs (*Figure 11B*). It is also not expressed in the posterior group of type II NBs but only marks type II NBs of the anterior-medial group. Finally, we found that *Tc-six3* marks, and is restricted to, the anterior-most lineages 1–4, including both the type II NBs and the fez-mm-eGFP marked INPs/GMCs (*Figure 11C*). These results reveal candidates for distinguishing type II identity from type I NBs in the brain. Only the latter is possibly marked by *Tc-otd* as *Drosophila otd* is expressed in type I brain NBs (*Urbach and Technau, 2004*). By contrast, *Tc-six4* only marks type II NBs and appears to be an early determinant of the anterior-medial group. *Tc-six4* is also a candidate for distinguishing the anterior-medial from the posterior cluster. The results also reveal a subdivision of the anterior cluster by *Tc-six3*, which marks the four most anterior type II NBs.

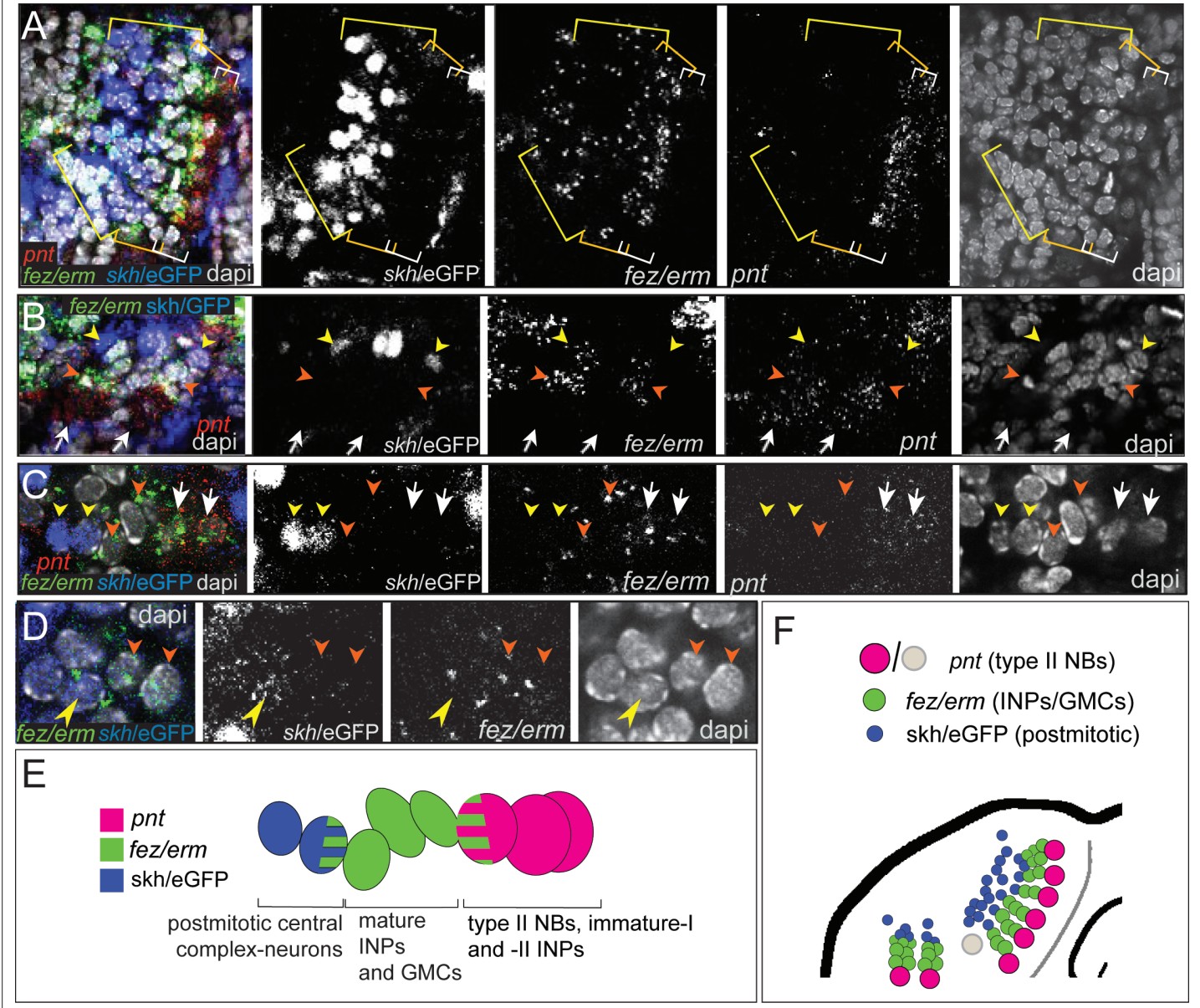

**Figure 8.** Expression of the transgenic central complex reporter skh/eGFP in relation to type II neuroblast (NB) lineages. (**A–D**) Antibody labeling of eGFP, which is expressed in the pattern of skh (*Garcia-Perez et al., 2021*) in combination with HCR visualizing *Tc-fez/erm* and *Tc-pnt*, DAPI labeling of nuclei, single planes of confocal z-stacks, all stage NS14. (**A**) Left head lobe, anterior group of *Tc-pnt*-positive type II NBs, *Tc-fez/erm*-expressing intermediate progenitors (INPs)/ganglion mother cells (GMCs) and skh/eGFP-positive postmitotic cells. (**B**) Left head lobe, posterior group of type II NB lineages. *Tc-pnt+* type II NBs or immature-I/-II INPs (white arrows), *Tc-fez/erm+* INPs/GMCs (orange arrowheads) and skh/eGFP postmitotic cells (yellow arrowheads). (**C**) Detail of one lineage including type II NB cluster (*Tc-pnt+*, white arrows), INPs/GMCs (*fez+*, orange arrowheads), and skh/eGFP cells at the end of the lineage (yellow arrowheads). (**D**) Detail of transition from *Tc-fez/erm+* cells (orange arrowheads) to skh/eGFP cells reveals a small area of co-expression of both factors (yellow arrowhead). (**E**) Schematic drawing of type II NB lineage showing relative positions and markers of type II NBs, INPs/GMCs, and postmitotic central complex forming cells. (**F**) Schematic overview of the arrangement of the type II NB lineages including skh/eGFP+ central complex forming cells in the head lobe of stage NS14 (immature-I/-II INPs not shown for simplicity). No data on skh-GFP expression in lineage of type II NB 7 (marked in gray) is available.

## Discussion

### Evolutionary divergence of number and grouping of embryonic type II NB lineages between beetle, fly, and grasshopper

We have identified a total of nine type II NB lineages on each side of the *Tribolium* embryonic head.

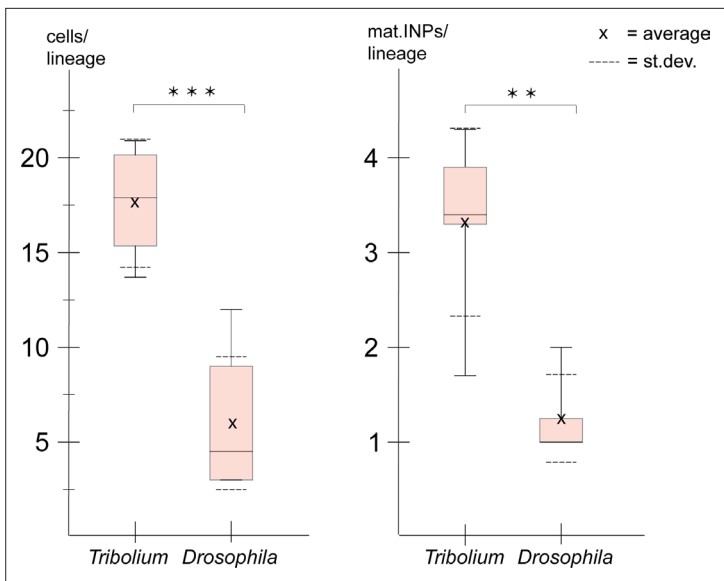

**Figure 9.** Size comparison of *Tribolium* and *Drosophila* type II neuroblast (NB) lineages. Anteriormedian lineages from *Tribolium* stages 13 and 14 (quantified in this work, see **Tables 1 and 2**) compared to *Drosophila* stages 15 and 16 (pooled data from **Walsh and Doe, 2017**). Left: lineage sizes comprising type II NBs, intermediate progenitors (INPs), and ganglion mother cells (GMCs) (*Tribolium* n=4; *Drosophila* n=8). Right: number of *Tc-dpn*-positive mature INPs (*Tribolium* n=5; *Drosophila* n=8). *t*-test p-value significance level (*t*-test; *** ≙p<0.001, ** ≙p<0.01).

This represents a major evolutionary divergence as in both the hemimetabolan grasshopper *S. gregaria* and the fruit fly *D. melanogaster* only eight embryonic type II NBs were identified (**Walsh and Doe, 2017**; **Boyan and Williams, 2011**). Our finding of an additional type II NB lineage at the embryonic stage shows that the number of type II NBs is not fixed across insects and suggests that

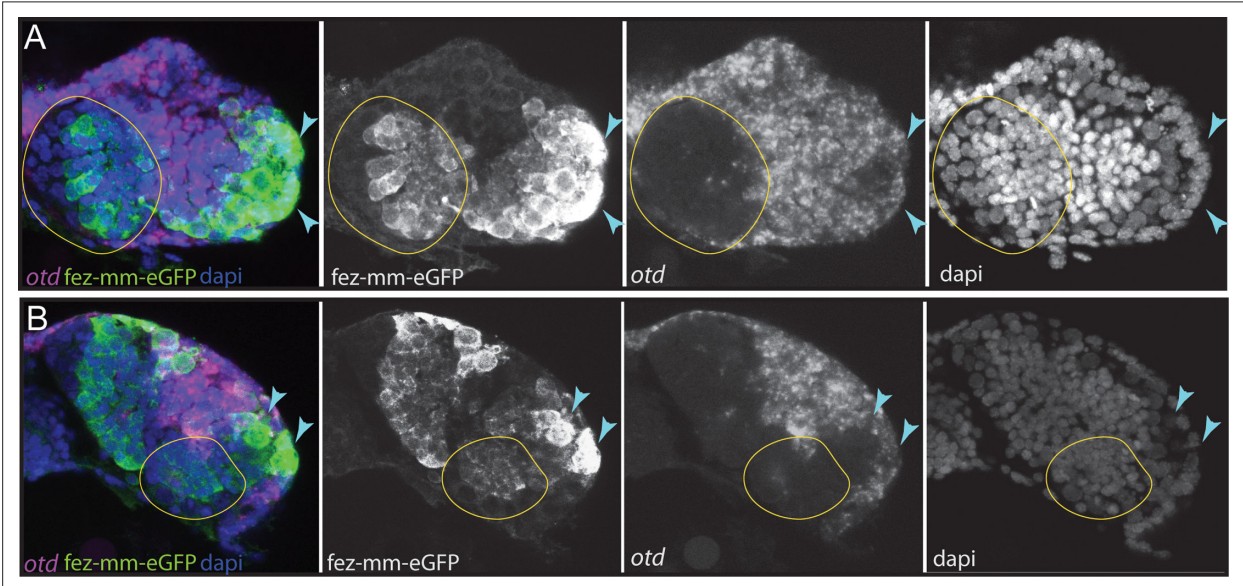

**Figure 10.** Expression of *Tc-otd* in relation to the fez-mm-eGFP lineages. (**A/B**) RNA in situ hybridization for *Tc-otd* in combination with GFP antibody staining and DAPI labeling of nuclei. (**A**) and (**B**) show projections of different levels of NS14 embryos, right head lobe. (**A**) *Tc-otd* is neither expressed in type II neuroblasts (NBs) nor the in the fez-mm-eGFP-positive intermediate progenitors (INPs) of the anterior-medial group (both positioned within encircled area) but it is expressed in the directly surrounding tissue. (**B**) Likewise, *Tc-otd* is not expressed in type II NBs or fez-mm-eGFP cells of the posterior group (encircled area). A lateral area of *Tc-fez/erm* and *Tc-otd* co-expression (**A/B**, blue arrow) likely is part of the eye anlagen (**Posnien et al., 2011b**).

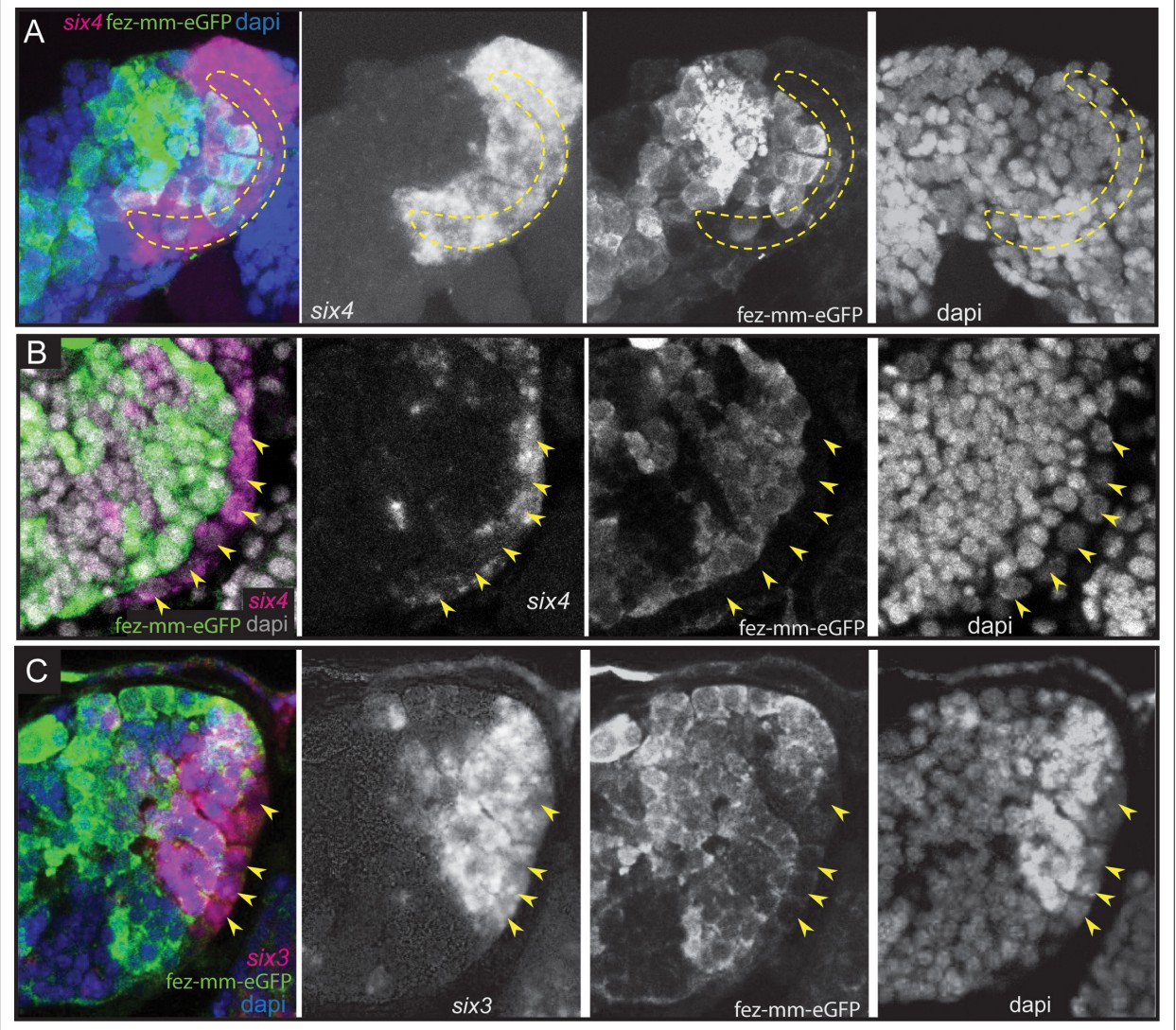

**Figure 11.** Expression of *Tc-six4* and *Tc-six3* in type II neuroblasts (NBs) and intermediate progenitors (INPs). (**A–C**) RNA in situ hybridizations of *Tc-six4* and *Tc-six3* in combination with GFP antibody staining reflecting fez-mm-eGFP expression, DAPI labeling of nuclei, left head lobes shown. (**A**) Stage NS13; *Tc-six4* is expressed in an area encompassing type II NBs (positioned in encircled area) and mm-eGFP expressing INPs. (**B**) Stage NS14; *Tc-six4* is now restricted to type II NBs and their youngest progeny that do not express fez-mm-eGFP (yellow arrowheads). (**C**) Stage NS14; *Tc-six3* specifically marks the four anterior-most lineages. It is expressed in the type II NBs at the base of the lineages (yellow arrowheads) and in the fez-mm-eGFP-positive INPs and possibly also in GMCs. It is absent from the remaining lineages of the anterior-medial group and is also not seen in the posterior group (not shown).

evolutionary divergence in brain development between species may be driven by differences in the number of progenitor lineages. To test this, further research is required to determine where the additional lineage in *Tribolium* is located and if it is associated with morphological differences of the brain.

Although in the frame of this work we did not perform lineage tracing, we can speculate about which NB might be lacking in fly embryos based on our data. In the hemimetabolan *Schistocerca*, there is no obvious clustering of type II NBs, and they are all arranged in one row at the medial rim of the head lobes. In *Tribolium*, we have observed an arrangement of the type II NBs into two groups per side, one large anterior-medial group containing seven type II NB clusters and one posterior group of two type II NBs. The anterior six of the anterior-medial group are arranged in one row like in the grasshopper. In *Drosophila*, there are three described groups referred to as the anterior, the middle, and the posterior clusters (**Walsh and Doe, 2017**). The posterior cluster consists of two type II NBs and therefore most likely corresponds to the posterior two type II NBs in *Tribolium*. The *Drosophila* middle

and anterior clusters are most likely equivalent to the anterior-medial group of *Tribolium*. However, the type II NB 7, which we assigned to the anterior-medial group, but which is a bit separated and produces offspring into the opposite direction (see *Figure 2C-II*, white arrow) might be the one that does not have a homologue in the fly embryo. The identification of more specific spatial markers for type II NBs (in addition to *Tc-six3* and *Tc-six4*) or lineage-tracing tools are required to identify the NB, which is not present in fly embryos. Subsequently, it would be especially interesting to see what the role of the additional type II NBs in the beetle is and which neuropile it contributes to.

In summary, there is a tendency toward grouping of type II NBs into subsets in the holometabolan models that was not observed in the only hemimetabolan studied so far. A ninth type II NB was only found in the *Tribolium* embryo that represents a striking developmentally divergent feature, but its role and the homologization of individual type II NBs between the different insects require further studies.

## Gene expression identifies homologous cell types and suggests conservation of gene function between fly and beetle type II NB lineages

The genes defining the different stages of differentiation in type II NB lineages have been identified and intensively studied in *Drosophila*, but it had remained unclear in how far the respective patterns

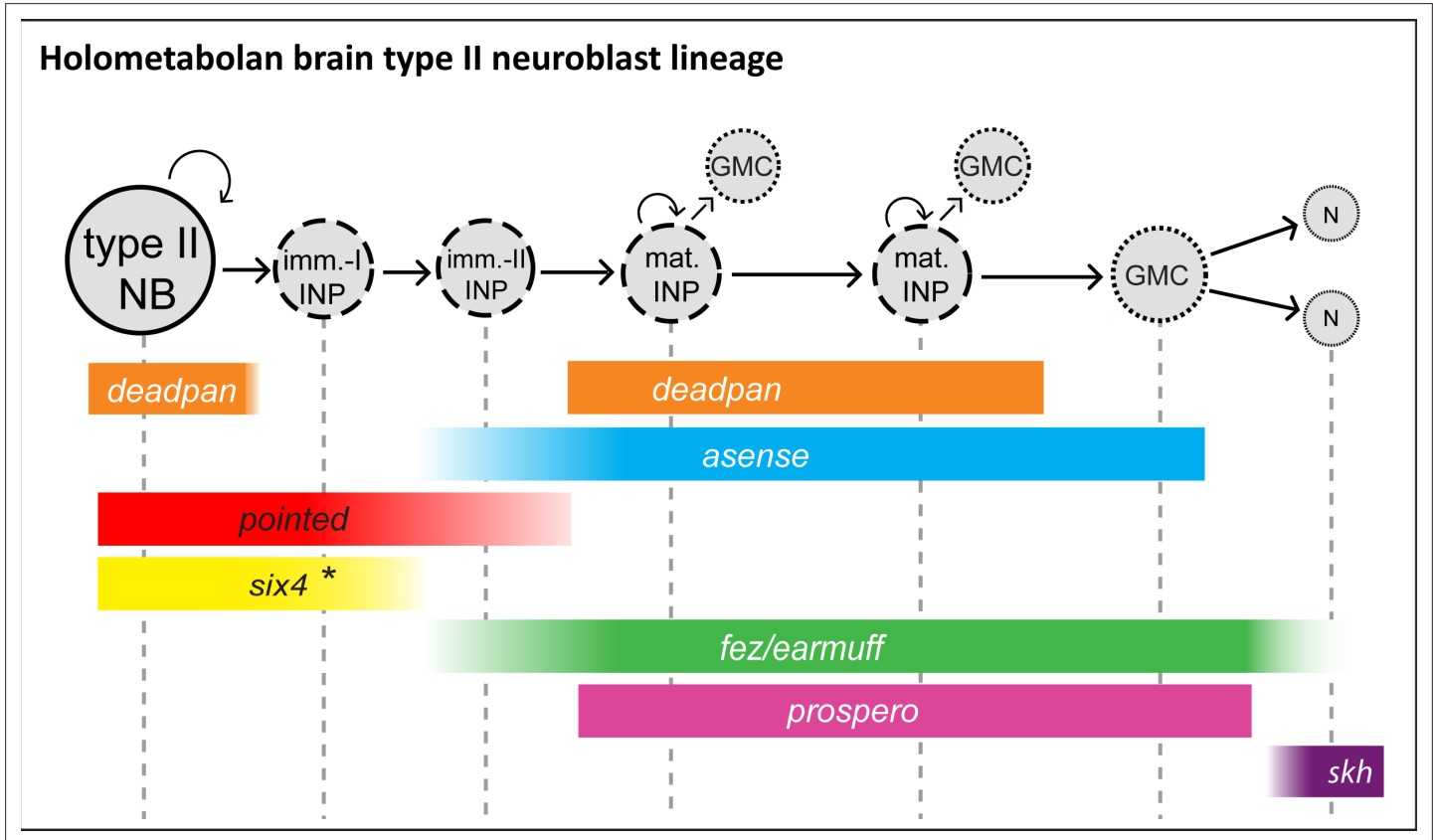

**Figure 12.** Conserved aspects of gene expression in type II neuroblast (NB) lineages of the holometabolan models *Drosophila* and *Tribolium*. Overview on the different cell types within one lineage (type II NBs, intermediate progenitors [INPs], ganglion mother cells [GMCs], and central complex neurons), their mitotic activity, and summary of the expression of key factors in the respective cell types. (I) NPs are divided into immature-I, immature-II, and mature INPs, with each subtype having a unique expression profile. Note that each lineage can have several mature INPs. We strongly assume that glia are also produced from these lineages in *Tribolium*, as they are in *Drosophila* (**Bayraktar and Doe, 2013**). * In the *Tribolium* embryo *Tc-six4* is only expressed in the anterior-medial lineages, and its expression is overlapping with *Tc-fez/erm* at an earlier stage (NS12), whereas it is restricted to type II NBs and immature-I INPs, mutually exclusive with *Tc-fez/erm*, at a later embryonic stage (NS14). *Drosophila six4* was found in all eight larval lineages (**Chen et al., 2021**). The scheme is based on this work and **Bayraktar and Doe, 2013**; **Garcia-Perez et al., 2021**; **Weng et al., 2010**; **Xie et al., 2016**; **Chen et al., 2021**. imm.-I/-II INP = immature-I/-II intermediate progenitor; mat. N=neuron; skh = shaking hands (enhancer trap) (**Garcia-Perez et al., 2021**).

and mechanisms were conserved. Our results show a large degree of conservation of expression and probably also function – at least in holometabola. We identified type II NBs by their larger-than-average size (**Boyan et al., 2010**) and the expression of the signature marker *Tc-pnt* adjacent to a group of *Tc-fez/erm*-expressing cells (INPs and GMCs). In *Drosophila*, both these factors have key functions in defining the developmental potential of type II NBs and INPs: Pnt suppresses Ase in type II NBs and promotes the formation of INPs (**Zhu et al., 2011**). Loss of *pnt* expression leads to a de-differentiation of INPs as *fez/erm* is no longer repressed in young INPs (**Xie et al., 2016**). Fez/Erm controls proliferation of INPs by activating Pros and prevents dedifferentiation of INPs into type II NBs (**Weng et al., 2010**). We show that also in *Tribolium* INPs undergo a maturation process, during which expression of *Tc-pnt* stalls and expression of *Tc-fez/erm* is switched on. Despite both factors characterizing different stages of INP maturation, they are not completely mutually exclusive and we observed a small window of overlap in immature-II INPs, highly suggestive of a conserved role of Pnt in the activation of *fez/erm* (**Xie et al., 2016**; **Figure 12**). *Drosophila dpn* is a neural marker (**Bier et al., 1992**) expressed in type II NBs and again in mature INPs, leaving a gap of expression in immature-I and II INPs. *Drosophila ase* is repressed in type II NBs but expressed together with *fez/erm* in INPs and GMCs where its expression overlaps with *pros* (**Li et al., 2016**). We found identical dynamics of gene expression in *Tribolium* (see **Figure 12**), suggesting conservation of gene function within the lineages and a conserved process of INP maturation.

In conclusion, expression of the genes examined in this study in type II NBs and their lineages is highly conserved in fly and beetle, which is implicit of conserved gene function and a conserved process of central complex formation.

## Divergent timing of type II NB activity and heterochronic development of the central complex

Previous work described a heterochronic shift in central complex formation between fly and beetle (**Farnworth et al., 2020b**; **Koniszewski et al., 2016**). In *Tribolium*, parts of the central complex develop during embryogenesis. It becomes functional at the onset of the first larval stage and is presumably required for the more complex movements of the beetle larvae using legs (**Farnworth et al., 2020b**). In both *Drosophila* and *Tribolium*, the anterior groups of type II NBs (anterior-medial group of *Tribolium* and anterior and middle cluster of *Drosophila*) are fully present in the last third of embryogenesis, with the beginning of germ band retraction (**Biffar and Stollewerk, 2014**; **Walsh and Doe, 2017**; **Campos-Ortega and Hartenstein, 1985**). However, in the *Drosophila* embryo, only a limited number of INPs and GMCs are produced before they enter a resting stage prior to hatching (**Walsh and Doe, 2017**). By contrast, we show that *Tribolium* type II NB lineages are much larger at the embryonic stage compared to *Drosophila* (**Walsh and Doe, 2017**). They also include more mature, *Tc-dpn*-expressing cycling INPs, which have the capability of driving the increase in lineage size. We believe that this increased activity may contribute to the embryonic development of a functional central body and protocerebral bridge (**Farnworth et al., 2020b**). This hypothesis does however require further confirmation by lineage-tracing experiments and testing how manipulating central genes acting within the lineages will affect the timing of central complex development. In addition, differences in cell number in the individual central complex neuropiles between *Tribolium* and *Drosophila* have to be assessed as the higher number of progenitors may also reflect a higher number of neurons present in the adult beetle fan-shaped body. Alternatively, a slower rate of division of the progenitor cells may result in their accumulation as they stay in their proliferative phase for longer while new progenitors are produced. Lastly, we do not yet have any information whether *Tribolium* type II NBs, or *Tribolium* NBs in general, also enter a stage of quiescence at the end of embryogenesis as they do in *Drosophila* (**Walsh and Doe, 2017**). It is also not known what role neurons born in the first larva play in *Tribolium* and if they contribute to the functionality of the larval central complex.

Another emerging question is if and when *Tribolium* type II NB lineages are active during larval development and how that relates to the described steps of central complex development. After preformation in the embryo *Drosophila*, type II NBs have their main period of activity in the third larva and the lineages are much larger and contain more INPs than the *Tribolium* embryonic lineages (**Bayraktar and Doe, 2013**). Therefore, it would be interesting to see if type II NB lineages are present and active in the late larval brain of *Tribolium*.

Taken together, we have for the first time identified and molecularly characterized embryonic type II NBs and INPs as the major central complex progenitor pool in an insect outside *Drosophila*. We have observed quantitative differences within these lineages when compared between *Tribolium* and *Drosophila*. Based on these findings, we hypothesize that increased division behavior of type II NB lineages is required for the embryonic central complex development as found in beetles. Our findings set the basis for future work on the relationship of the identified type II NB lineages and the timing of central complex development.

## The placodal marker *Tc-six4* may be responsible for the differentiation and the spatial identity of the anterior-medial group of type II NBs

The different type II NB lineages contribute to different parts of the brain. For instance, only the anterior four type II NB lineages form the w, x, y, and z tracts of the central complex in flies and grasshoppers. Hence, there must be signals that make them different despite the conserved and well-studied sequence of neural gene expression and functional interactions in *Drosophila* (*Figure 12*; *Bayraktar and Doe, 2013*; *Weng et al., 2010*; *Xie et al., 2016*; *Li et al., 2016*). In the ventral nerve cord, the different identities of the NBs are specified by the combination of spatial patterning genes expressed in the neuroectoderm at the time of delamination (*Stollewerk, 2016*). However, little is known about these signals for type II NBs. In *Tribolium*, the domains of head patterning genes in the brain neuroectoderm are well characterized and some highly conserved territories like the anterior-medial *six3*-positive domain were described and suggested to give respective NBs different spatial identities (*Posnien et al., 2023*; *Posnien et al., 2011b*; *Steinmetz et al., 2010*). The *Tc-six4* expression domain is also largely coincident with an embryonic structure termed insect head placode, a neurogenic, invaginating tissue (*Posnien et al., 2011a*; *de Velasco et al., 2007*) potentially homologous to the vertebrate adenohypophyseal placode, which is also marked by *six4* (*Posnien et al., 2023*; *Posnien et al., 2011a*). A placodal origin of *Drosophila* type II NBs has been shown for one type II NB, and it is assumed that the other type II NBs also stem from placodal tissue (*Walsh and Doe, 2017*; *Álvarez and Díaz-Benjumea, 2018*; *Hwang and Rulifson, 2011*).

*Tc-six4* is also a very interesting factor with regards to the specification of type II NBs because it is expressed only in the anterior group of type II NBs. Hence, it could contribute to distinguishing their developmental fate from the posterior group. This is however different in the *Drosophila* larva. Here, Six4 is expressed in all eight lineages where it prevents the formation of supernumerary type II NBs and a premature differentiation of INPs (*Chen et al., 2021*). In the *Tribolium* embryo Six4 may have a similar rate-limiting role within the anterior group of type II NBs. However, its early expression in a small part of the neuroectoderm (*Posnien et al., 2011a*) and the delamination of the anterior-medial type II NB group within this domain also hints at an instructive role of *Tc-six4* in the formation of the anterior-medial group.

## *Tc-six3* marks a subset of type II NBs, whereas *Tc-otd* is absent from all lineages

Importantly, we found that in *Tribolium* late embryogenesis, *Tc-six3* is expressed specifically within the lineages of type II NBs 1–4 of the anterior-medial group, but not in the other lineages. In grasshopper and fly, these anterior 4 lineages (DM1-4) give rise to the z, y, x, and w tracts (*Boyan and Reichert, 2011*; *Boyan and Williams, 2011*) and contribute crucially to the development of protocerebral bridge and central body. These tracts form a major midline crossing neuropile of the central body, which serves as a scaffold in the development of this structure (*Farnworth et al., 2020b*; *Boyan and Reichert, 2011*). We conclude that in *Tribolium* the evolutionary ancient *six3* territory gives rise to the neuropile of the z, y, x, and w tracts. This might well be the same in other insects such as flies and grasshoppers but has to our knowledge not been looked at in detail in these models. In the beetle, the central body is also missing in weak *Tribolium Tc-six3*-RNAi phenotypes, suggesting that *Tribolium six3* is required for the formation of this structure (*Posnien et al., 2011b*). Given that central complex neuropile is marked by additional factors such as *Tc-foxQ2* (*Drosophila* fd102c; *Lee and Frasch, 2004*) and *Tc-rx-* (*Farnworth et al., 2020b*; *He et al., 2019*), it would be interesting to see the relationship of additional spatial patterning genes with regards to the type II NB lineages in future studies.

The gene *otd* is a marker of the posterior protocerebrum, and *six3* and *otd* are believed to subdivide the embryonic anterior brain into two major domains (*Steinmetz et al., 2010*). This subdivision

is also part of the recently revisited concept of an ancestral division of the insect protocerebrum into archicerebrum (*six3*-positive) and prosocerebrum (*otd*-positive) (***Posnien et al., 2023***). As discussed above, *Tc-six3* is expressed in the anterior-most four type II NB lineages, but unexpectedly, we found that *Tc-otd* is specifically absent from all type II NB lineages, suggesting that its inhibition is required for the development of all these lineages. Interestingly, the lineages of type II NB 5 and 6 (and presumably of type II NB 7) of the anterior cluster and the posterior type II NB lineages do neither express *Tc-otd* nor *Tc-six3*, showing that there are protocerebral structures expressing none of these conserved markers in development, questioning the assigned generalized role of these two factors in subdividing the protocerebrum (***Posnien et al., 2023***; ***Steinmetz et al., 2010***).

In summary, our findings are just the beginning of the quest for the genes required for the identity specification of type II NB lineages. The genes known to be involved in head patterning are an excellent starting point for this purpose (***Posnien et al., 2023***; ***Posnien et al., 2010***).

# Materials and methods

## Key resources table

| Reagent type (species) or resource | Designation | Source or reference | Identifiers | Additional information |
|---|---|---|---|---|
| Strain, strain background (*Tribolium castaneum*) | vermillion*white* | Bucher lab stock, University of Göttingen | *vermillion-white* (*vw*) | For transgenesis, mutant eye color (white) is rescued to black by 3XP3-vw |
| Genetic reagent (*T. castaneum*) | skh GFP enhancer trap line | ***Garcia-Perez et al., 2021***, Bucher lab stock, University of Göttingen | G10011-GFP | Central complex reporter line |
| Genetic reagent (*T. castaneum*) | fez/erm GFP enhancer trap line | This paper, Bucher lab stock, University of Göttingen | fez-mm-eGFP | New CRISPR-Cas9 generated line analyzed in this work |
| Recombinant DNA reagent | [3xP3:Tc'v-SV40-Cre-2A-EGFP:bhsp68-eb] | ***He et al., 2019***, Addgene plasmid #124068 | | NHEJ repair template |
| Recombinant DNA reagent | [bhsp68-Cas9] | ***Gilles et al., 2015***, Addgene plasmid #65959 | Cas9 helper-plasmid | |
| Recombinant DNA reagent | [U6b-BsaI-gRNA] | ***Gilles et al., 2015***, Addgene plasmid #65956 | *Tribolium* U6b promoter with BsaI cloning site to insert guide RNA sequence | |
| Sequence-based reagent | *Tc-fez/erm* upstream 1 | This paper | CRISPR guide RNA | GTGATTACGTGCCGCCGAAG |
| Sequence-based reagent | *Tc-fez/erm* upstream 2 | This paper | CRISPR guide RNA | GCGCTTGCTCGGTTCTCAGT |
| Sequence-based reagent | *Tc-fez/erm* upstream 3 | This paper | CRISPR guide RNA | GCCGTCGTGAGTGAAACGCC |
| Sequence-based reagent | *Dm-ebony* | ***He et al., 2019*** | CRISPR guide RNA | GAACCGGGCAGCCCGCCTCC |
| Sequence-based reagent | *Dm-yellow* | ***He et al., 2019*** | CRISPR guide RNA | GCGATATAGTTGGAGCCAGC |
| Sequence-based reagent | GFP-5'-rv1 | This paper | Primer | TGAACTTGTGGCCGTTTACG |
| Sequence-based reagent | fez-exon1-rv1 | This paper | Primer | AACATTAGGTGAGCAGGGCC |
| Sequence-based reagent | P2A-rv | This paper | Primer | TCTTCCACGTCTCCTGCTTG |

*Continued on next page*

*Continued*

| Reagent type (species) or resource | Designation | Source or reference | Identifiers | Additional information |
|---|---|---|---|---|
| Sequence-based reagent | GFP-fw-1 | This paper | Primer | TTCTTCAAGGACGACGGCAA |
| Sequence-based reagent | Cre-rv1 | This paper | Primer | GTTGCATCGACCGGTAATGC |
| Sequence-based reagent | *Tc-fez/earmuff* (TC004673) RNA probe | *Posnien et al., 2011b* | Gene-specific forward and reverse primers for probe template | fw:CAAGCCCTCCATCGTGACCC rv:GAATCGGA GGCGGAAGTACT |
| Sequence-based reagent | *Tc-pointed* (TC034783) RNA probe | This paper | Gene-specific forward and reverse primers for probe template | fw:GACCGCTTTATTTGCATTGT rv:TGCTTCTC GTAGTTCATCTTC |
| Sequence-based reagent | *Tc-asense* (TC008437) RNA probe | *Posnien et al., 2011b* | Gene-specific forward and reverse primers for probe template | fw:CGTCAGTGTGGTATCCCCTC rv:GCTGTTCC CACCACTGCATGA |
| Sequence-based reagent | *Tc-deadpan* (TC005224) RNA probe | This paper | Gene-specific forward and reverse primers for probe template | fw:CTCGAGTAACTTACCATTT rv:CTACCACG GCCTCCACATA |
| Sequence-based reagent | *Tc-prospero,* (TC010596) RNA probe | This paper | Gene-specific forward and reverse primers for probe template | fw:ACACAGGGATTCTCGGTCTC rv:TTGCGTGT CCAAGCAAGAA |
| Sequence-based reagent | *Tc-six4,* (TC003852) RNA probe | *Posnien et al., 2011a* | Gene-specific forward and reverse primers for probe template | fw:AAGTCGGCGCGAAAGAACGG rv:CTAAATTT ATGGTACTTGAT |
| Sequence-based reagent | *TC-six3,* (TC000361) RNA probe | *Posnien et al., 2011b* | Gene-specific forward and reverse primers for probe template | fw:ATATGGCGCTCGGACTCGGC rv:CTCGTACG GTATATATCACG |
| Sequence-based reagent | *Tc-otd,* (TC003354) RNA probe | *Posnien et al., 2011b* | Gene-specific forward and reverse primers for probe template | fw:ATGTGGCCTCCAGAGGCAGT rv:TTAAGCCATATTTGCAAACT |
| Sequence-based reagent | *Tc-fez/erm* (TC004673) *label B1 probe* | Molecular Instruments | Probe for hybridization chain reaction | |
| Sequence-based reagent | *Tc-deadpan* (TC005224) *label B2 probe* | Molecular Instruments | Probe for hybridization chain reaction | |
| Sequence-based reagent | *Tc-asense* (TC008437) *label B3 probe* | Molecular Instruments | Probe for hybridization chain reaction | |
| Sequence-based reagent | *Tc-pointed* (TC034783) *label B4 probe* | Molecular Instruments | Probe for hybridization chain reaction | |
| Sequence-based reagent | *B1-* Alexa Fluor 488, 546 | Molecular Instruments | Hairpin | |
| Sequence-based reagent | *B2-* Alexa Fluor 456, 514 | Molecular Instruments | Hairpin | |
| Sequence-based reagent | *B3-* Alexa Fluor 647, 488 | Molecular Instruments | Hairpin | |
| Sequence-based reagent | *B4-* Alexa Fluor 594, 647 | Molecular Instruments | Hairpin | |
| Antibody | Anti-GFP primary antibody (chicken polyclonal) | Abcam | ab13970 | Dilution: 1/1000 v/v |
| Antibody | Anti phospho-histone 3 primary antibody (rabbit polyclonal) | Sigma-Aldrich | H0412 | Dilution: 1/100 v/v |

*Continued on next page*

*Continued*

| Reagent type (species) or resource | Designation | Source or reference | Identifiers | Additional information |
|---|---|---|---|---|
| Antibody | Secondary antibody coupled with Alexa Fluor 488 (goat anti-chicken, polyclonal) | Thermo Fisher Scientific | A-11039 | Dilution: 1/1000 v/v |
| Antibody | Secondary antibody coupled with Alexa Fluor 647 (goat anti-rabbit, polyclonal) | Thermo Fisher Scientific | A-21244 | Dilution: 1/500 v/v |
| Antibody | Anti-Digoxigenin-POD (poly), Fab fragments (sheep polyclonal) | Roche | 11633716001 | Dilution: 1/2000 v/v |
| Antibody | Anti-Fluorescein-POD, Fab fragments (sheep polyclonal) | Roche | 11426346910 | Dilution: 1/2000 v/v |
| Chemical compound, drug | Tyramide conjugate Alexa Fluor 555 | Invitrogen | B40955 | Dilution: 1/250 v/v |
| Chemical compound, drug | DAPI nuclear dye | Invitrogen | D1306 | Dilution: 1/1000 (of 1 µg/µl stock sol.) |
| Chemical compound, drug | FM1-43 | Invitrogen | T3163 | Dilution: 1/1000 (of 5 µg/µl stock sol.) |
| Chemical compound, drug | VECTASHIELD Antifade Mounting Medium | Vector Laboratories | VEC-H-1000 | |

## Creation of transgenic reporter line

Using CRISPR-Cas9, we have inserted a transgene into the *Tc-fez/erm* locus that includes *egfp* behind the *Tribolium* basal heat shock promoter (bhsp68), which is not active by itself but can be activated by nearby enhancer elements (*Schinko et al., 2010*; *Schinko et al., 2012*). The transgene also contained the coding sequence of Cre-recombinase (not used in this work) transcribed from the same promoter, and the eye pigmentation gene *Tc-vermilion* behind the eye-specific 3xP3 enhancer element (*Berghammer et al., 1999*; see *Figure 2—figure supplement 1A* for map of the transgene). On the repair plasmid, the transgene was flanked by CRISPR target sequences derived from *Dm-yellow* and *Dm-ebony*, which were used for in vivo linearization of the transgene.

Using CRISPR Optimal Target Finder (*Gratz et al., 2014*), we designed three guide RNAs (gRNAs) targeting the genomic region 2.5 kb upstream of the *Tc-fez/erm* transcription start site (see *Supplementary file 1a*). We produced three plasmids where single gRNAs were transcribed from the *Tribolium* pU6b RNA promoter, as described in *Gilles et al., 2015*. We co-injected the three gRNAs plasmids at 125 µg/µl each, gRNAs targeting the *Dm-yellow* and *Dm-ebony* sequence at 125 µg/µl each, Cas9 helper plasmid (see *Gilles et al., 2015*) at 500 µg/µl and the repair plasmid at 500 µg/µl. Embryos of the white-eyed *Tribolium vermilion-white* (*vw*) strain (G0) were injected, raised, and then bred to *vw*-beetles in individual crosses. The offspring of these crosses (F1) were screened for black-eyed heterozygous carriers of the transgene. These were again outcrossed to *vw*-beetles and the heterozygous offspring (F2) were further analyzed genetically (see *Supplementary file 1b*) and with respect to fluorescent signal. Siblings of the F2 generation were then crossed to one another to generate the homozygous line (fez-mm-eGFP). Details of the insertion can be found in *Figure 2—figure supplement 1*.

We also used the shaking hands (skh) enhancer trap line (G10011-GFP) which marks central complex cells (*Garcia-Perez et al., 2021*).

## Larval brain dissection, fixation, and staining

Larval brains of the newly generated *Tc-fez/erm* enhancer trap line fez-mm-eGFP were dissected and stained as described in *Hunnekuhl et al., 2020*. Brains were mounted in *Vectashield* for microscopic inspection. See the 'Key resources table' for antibodies and staining reagents that were used.

## Embryonic RNA in situ hybridization, hybridization chain reaction, and antibody staining

Gene identifiers and primer sequences that were used for amplification can be found in the 'Key resources table'. The fragments were inserted into a pJet1.2 cloning vector. Standard RNA in situ probes were synthesized using the 5X Megascript T7 kit (Ambion) according to the manufacturer's protocol. Embryo fixation and single-color RNA in situ chain reaction in combination with antibody staining to eGFP or phospo-histone H3 staining to mark mitoses was conducted as described in *Buescher et al., 2020*. Embryos were also stained for DNA (DAPI) or cell membranes (FM1-43). All antibodies and staining reagents are listed in the 'Key resources table'. Probes for multicolor *HCR* binding *Tc-fez/erm*, *Tc-dpn*, *Tc-ase,* and *Tc-pnt* were produced by Molecular Instruments (see the 'Key resources table'). Labeling reactions were performed as described in *Tidswell et al., 2021*. Antibody staining to eGFP was performed following the completion of the HCR staining (see *Buescher et al., 2020*). All embryos were mounted in *Vectashield* for microscopic inspection (see the 'Key resources table').

## Image acquisition and analysis

Multichannel image stacks were recorded using a Zeiss LSM 980 confocal laser scanning microscope. Image stacks consisted of 100–300 slices, depending on the specimen. The resolution ranged from 1024 × 1024 to 2048 × 2048 pixels. Plane thickness was optimized according to the width of the pinhole and ranged from 0.1 to 1 µm. We used FIJI (*Schindelin et al., 2012*) for three-dimensional inspection to export individual planes or to generate maximum intensity projections. Cropping and adjustment of brightness and contrast were done with GIMP (version 2.10.32) or Adobe Photoshop CS5.

## Evaluation of cell size and cell number

Cell and nuclear diameters of type II NBs and a control group were measured based on the image stacks using FIJI (*Schindelin et al., 2012*) across seven embryos. Lineage size and numbers of mitoses (based on *Tc-pnt* and *fez*-mm-eGFP expression/anti-PH3 staining) were evaluated by manual counting based on image stacks derived from four head lobes of four embryos (≙ 28 anterior-medial lineages). The number of *Tc-dpn* expressing INPs was evaluated in the same way across five embryos (≙ 35 anterior-medial lineages). Reference data on *Drosophila* lineage size (type II NBs, INPs, GMCs, and neurons) and numbers of *dpn*-expressing intermediate INPs were extracted from *Walsh and Doe, 2017*. Statistics on all numerical data (SDs and two-tailed *t*-tests) were performed using Microsoft Excel 2016.

## Acknowledgements

We thank Claudia Hinners for technical support with molecular biology methods and Elke Küster for help with screening and beetle stock keeping. We also thank Christoph Viehbahn for valuable feedback on the project. SR was supported by a scholarship of the *Göttingen Promotionskolleg für Medizinstudierende*, funded by the *Jacob-Henle-Programm*. We acknowledge support by *Niedersachsen Open* and the *Open Access Publication Funds/transformative agreements* of the Göttingen University.

## Additional information

### Funding

| Funder | Grant reference number | Author |
| --- | --- | --- |
| Göttingen Promotionskolleg für Medizinstudierende | Jacob-Henle-Programm | Simon Rethemeier |

The funders had no role in study design, data collection and interpretation, or the decision to submit the work for publication.

## Author contributions
Simon Rethemeier, Data curation, Investigation, Visualization, Writing - review and editing; Sonja Fritzsche, Dominik Mühlen, Data curation, Investigation; Gregor Bucher, Supervision, Writing - review and editing; Vera S Hunnekuhl, Conceptualization, Data curation, Supervision, Investigation, Visualization, Writing - original draft, Writing - review and editing

## Author ORCIDs
Simon Rethemeier ⬤ https://orcid.org/0009-0007-6990-7961
Sonja Fritzsche ⬤ https://orcid.org/0000-0003-3335-3534
Dominik Mühlen ⬤ http://orcid.org/0000-0002-0530-9360
Gregor Bucher ⬤ https://orcid.org/0000-0002-4615-6401
Vera S Hunnekuhl ⬤ https://orcid.org/0000-0002-9100-2161

Reviewer #1 (Public review): https://doi.org/10.7554/eLife.99717.3.sa1
Reviewer #2 (Public review): https://doi.org/10.7554/eLife.99717.3.sa2
Reviewer #3 (Public review): https://doi.org/10.7554/eLife.99717.3.sa3
Author response https://doi.org/10.7554/eLife.99717.3.sa4

## Additional files

### Supplementary files
Supplementary file 1. CRISPR guide RNA and primer sequences. (a) Target sites for CRISPR-Cas9-mediated *non-homologous end joining* knock-in of the eGFP containing transgene (see *Figure 2—figure supplement 1A*). Guide sequences 1–3 are within 2.6 kb upstream of the transcription start site (TSS). (b) Primer sequences for line fez-mm-eGFP insertion site testing. See *Figure 2—figure supplement 1A* for primer binding sites.

MDAR checklist

### Data availability
The data (confocal image stacks) that were analysed for this study are available at https://doi.org/10.25625/8IVICL.

The following dataset was generated:

| Author(s) | Year | Dataset title | Dataset URL | Database and Identifier |
|---|---|---|---|---|
| Rethemeier S, Fritzsche S, Mühlen D, Bucher G, Hunnekuhl VS | 2024 | Data accompanying the manuscript 'Differences in size and number of embryonic type II neuroblast lineages are associated with divergent timing of central complex development between beetle and fly' | https://doi.org/10.25625/8IVICL | Göttingen Research online data, 10.25625/8IVICL |

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
