## [Editor Report · eLife Assessment]

The study is a **valuable** contribution to the question of evolutionary shifts in neuronal proliferation patterns and the timing of developmental progressions. The authors present **convincing** data which confirm the presence of type II NB lineages in beetle with the same molecular characteristics as the *Drosophila* counterparts but differing in lineage size and number. The data lay the foundation for future analysis of the role and molecular characteristics of individual lineages and of whether differences in the identity, proliferation pattern, and timing of developmental progression can be linked to differences in the development of functionality of the central complex.

---

## [Referee Report · Reviewer #1 (Public review)]

Summary:

Insects inhabit diverse environments and have neuroanatomical structures appropriate to each habitat. Although the molecular mechanism of insect neural development has been mainly studied in *Drosophila*, the beetle, Tribolium castaneum has been introduced as another model to understand the differences and similarities in the process of insect neural development. In this manuscript, the authors focused on the origin of the central complex. In *Drosophila*, type II neuroblasts have been known as the origin of the central complex. Then, the authors tried to identify those cells in the beetle brain. They established a Tribolium fez enhancer trap line to visualize putative type II neuroblasts and successfully identified 9 of those cells. In addition, they also examined expression patterns of several genes that are known to be expressed in the type II neuroblasts or their lineage in *Drosophila*. They concluded that the putative type II neuroblasts they identified were type II neuroblasts because those cells showed characteristics of type II neuroblasts in terms of genetic codes, cell diameter, and cell lineage.

Strengths:

The authors established a useful enhancer trap line to visualize type II neuroblasts in Tribolium embryos. Using this tool, they have identified that there are 9 type II neuroblasts in the brain hemisphere during embryonic development. Since the enhancer trap line also visualized the lineage of those cells, the authors found that the lineage size of the type II neuroblasts in the beetle is larger than that in the fly. They also showed that several genetic markers are also expressed in the type II neuroblasts and their lineages as observed in *Drosophila*.

Comments on revisions:

The revisions have improved the manuscript greatly. However, I still have some concerns about the lack of examination of the expression of NB markers. Without examining the expression of at least one unequivocal neuroblast marker, no one can say confidently that it is a neuroblast. However, it is acknowledged that such a marker is currently not available for Tribolium.

---

## [Referee Report · Reviewer #2 (Public review)]

The authors address the question of differences in the development of the central complex (Cx), a brain structure mainly controlling spatial orientation and locomotion in insects, which can be traced back to the neuroblast lineages that produce the Cx structure. The lineages are called type-II neuroblast (NB) lineages and assumed to be conserved in insects. While Tribolium castaneum produces a functional larval Cx that only consists of one part of the adult Cx structure, the fan-shaped body, in *Drosophila melanogaster* a non-functional neuropile primordium is formed by neurons produced by the embryonic type-II NBs which then enter a dormant state and continue development in late larval and pupal stages.

The authors present a meticulous study demonstrating that type-II neuroblast (NB) lineages are indeed present in the developing brain of Tribolium castaneum. In contrast to type-I NB lineages, type-II NBs produce additional intermediate progenitors. The authors generate a fluorescent enhancer trap line called fez/earmuff which prominently labels the mushroom bodies but also the intermediate progenitors (INPs) of the type-II NB lineages. This is convincingly demonstrated by high resolution images that show cellular staining next to large pointed labelled cells, a marker for type-II NBs in *Drosophila melanogaster*. Using these and other markers (e.g. deadpan, asense), the authors show that the cell type composition and embryonic development of the type-II NB lineages are similar to their counterparts in *Drosophila melanogaster*. Furthermore, the expression of the *Drosophila* type-II NB lineage markers six3 and six4 in subsets of the Tribolium type-II NB lineages (anterior 1-4 and 1-6 type-II NB lineages) and the expression of the Cx marker skh in the distal part of most of the lineages provide further evidence that the identified NB lineages are equivalent to the Drosophila lineages that establish the central complex. However, in contrast to Drosophila, there are 9 instead of 8 embryonic type-II NB lineages per brain hemisphere and the lineages contain more progenitor cells compared to the Drosophila lineages. The authors argue that the higher number of dividing progenitor cells supports the earlier development of a functional Cx in Tribolium.

While the manuscript clearly shows that type-II NB lineages similar to *Drosophila* exist in Tribolium, it does not establish a direct link between the characteristics of these lineages and a functional larval Cx in Tribolium, i.e., it does not identify the cause of the heterochronic development of the Cx in these insects. However, the detailed study lays the foundation for lineage tracing and gene function experiments that will elucidate if the higher number of Tribolium type-II NB lineage progenitors, the additional lineage and the timing of developmental progression of the progenitors can indeed be linked with the earlier function of the Cx and/or if other components are required for establishing the functional larval neural circuits in Tribolium such as e.g. larval born neurons as is the case in *Drosophila*.

---

## [Referee Report · Reviewer #3 (Public review)]

Summary:

In this paper, Rethemeier et al capitalize on their previous observation that the beetle central complex develops heterochronically compared to the fly and try to identify the developmental origin of this difference. For this reason, they use a fez enhancer trap line that they generated to study the neuronal stem cells (INPs) that give rise to the central complex. Using this line and staining against *Drosophila* type-II neuroblast markers, they elegantly dissect the number of developmental progression of the beetle type II neuroblasts. They show that the NBs, INPs, and GMCs have a conserved marker progression by comparing to *Drosophila* marker genes, although the expression of some of the lineage markers (otd, six3, and six4) is slightly different. Finally, they show that the beetle type II neuroblasts lineages are likely longer than the equivalent ones in *Drosophila* and argue that this might be the underlying reason for the observed heterochrony.

Strengths:

- Very interesting study system that compares a conserved structure that, however, develops in a heterochronic manner.

- Identification of a conserved molecular signature of type-II neuroblasts between beetles and flies. At the same time, identification of transcription factors expression differences in the neuroblasts, as well as identification of an extra neuroblast.

- Nice detailed experiments to describe the expression of conserved and divergent marker genes, including some lineaging looking into co-expression of progenitor (fez) and neuronal (skh) markers.

Weaknesses:

- The link between size and number of neuroblast lineages and the earlier central complex development in beetles is not examined.

---

## [Author Response]

The following is the authors’ response to the original reviews.

General Response to Public Reviews

We thank the three reviewers for their positive evaluation of our work, which presents the first molecular characterization of type-II NB lineages in an insect outside the fly *Drosophila.* They seem convinced of our finding of an additional type-II NB and increased proliferation during embryogenesis in the red flour beetle. The reviewers expressed hesitations on our interpretation that the observed quantitative differences of embryonic lineages can directly be linked to the embryonic development of the central complex in Tribolium. While we still believe that a connection of both observations is a valid and likely hypothesis, we acknowledge that due the lack of functional experiments and lineage tracing a causal link has not directly been shown. We have therefore changed the manuscript to an even more careful wording that on one hand describes the correlation between increased embryonic proliferation with the earlier development of the Cx but on the other hand also stresses the need for additional functional and lineage tracing experiments to test this hypothesis. We have also strengthened the discussion on alternative explanations of the increased lineage size and emphasize the less disputed elements like presence and conservation of type-II NB lineages.

While our manuscript could in conclusion not directly show that the reason of the heterochronic shift lies in the progenitor behaviour, we still provide a first approach to answering the question of the developmental basis of this shift and testable hypotheses directly emerge from our work. We agree with reviewer#1 that functional work is best suited to test our hypothesis and we are planning to do so. However, we believe that the presented work is already rich in novel data and significantly advances our understanding on the conservation and divergence of type-II NBs in insects. We would also like to stress that most transgenic tools for which genome-wide collections exist for *Drosophila* have to be created for *Tribolium* and doing so can be quite time consuming. Conducting RNAi experiments is certainly possible in Tribolium but observing phenotypes in this defined cellular context will need laborious optimization. We have for example tried knocking down Tc-fez/erm but could not see any embryonic phenotype which might be due to an escaper effect in which only mildly affected or wild type-like embryos survive while the others die in early embryogenesis. Due to pleiotropic functions of the involved genes a cell-specific knockdown might be necessary and we are working towards establishing a system to do that in the red flour beetle. For the stated reasons, we see our work as an important basis to inspire future functional studies that build up on the framework that we introduced.

In response to these common points, we have made the following changes to the manuscript

- The title has been changed from ‘being associated’ to ‘correlate’

- The conclusions part of the abstract has been changed

- We deleted the statement ‘…thus providing the material for the early central complex formation…’

- Rephrased to saying that the two observations just correlate

- The part of the discussion ‘Divergent timing of type-II NB activity and heterochronic development of the central complex’ has been extensively rewritten and now discusses several alternative explanations that were suggested by the reviewers. It also stresses the need for further functional work and lineage tracing (line 859-862 (608-611)).

In addition, we have made numerous changes to the manuscript to account for more specific comments of the reviewers and to the recommendations for the authors.

Our responses to the individual comments can be found in the following.

**Public Reviews:**

**Reviewer #1 (Public Review):**
Summary:Insects inhabit diverse environments and have neuroanatomical structures appropriate to each habitat. Although the molecular mechanism of insect neural development has been mainly studied in *Drosophila*, the beetle, Tribolium castaneum has been introduced as another model to understand the differences and similarities in the process of insect neural development. In this manuscript, the authors focused on the origin of the central complex. In *Drosophila*, type II neuroblasts have been known as the origin of the central complex. Then, the authors tried to identify those cells in the beetle brain. They established a Tribolium fez enhancer trap line to visualize putative type II neuroblasts and successfully identified 9 of those cells. In addition, they also examined expression patterns of several genes that are known to be expressed in the type II neuroblasts or their lineage in *Drosophila*. They concluded that the putative type II neuroblasts they identified were type II neuroblasts because those cells showed characteristics of type II neuroblasts in terms of genetic codes, cell diameter, and cell lineage.Strengths:The authors established a useful enhancer trap line to visualize type II neuroblasts in Tribolium embryos. Using this tool, they have identified that there are 9 type II neuroblasts in the brain hemisphere during embryonic development. Since the enhancer trap line also visualized the lineage of those cells, the authors found that the lineage size of the type II neuroblasts in the beetle is larger than that in the fly. They also showed that several genetic markers are also expressed in the type II neuroblasts and their lineages as observed in *Drosophila*.Weaknesses:I recommend the authors reconstruct the manuscript because several parts of the present version are not logical. For example, the author should first examine the expression of dpn, a well-known marker of neuroblast. Without examining the expression of at least one neuroblast marker, no one can say confidently that it is a neuroblast. The purpose of this study is to understand what makes neuroanatomical differences between insects which is appropriate to their habitats. To obtain clues to the question, I think, functional analyses are necessary as well as descriptive analyses.

The expression of an exclusive type-II neuroblast marker would indeed have been the most convincing evidence. However, *asense* is absent from type-II NBs and *deadpan* is not specific enough as it is expressed in many other cells of the developing protocerebrum. The gene *pointed*, although also expressed elsewhere, emerged as the the most specific marker. Therefore, we start with *pointed* and fez/erm to describe the first appearance and developmental progression of the cells and then add further evidence that these cells are indeed type-II neuroblasts. Further evidence is provided in the following chapters. We have discussed the need for functional work in the general response.

**Reviewer #2 (Public Review):**
The authors address the question of differences in the development of the central complex (Cx), a brain structure mainly controlling spatial orientation and locomotion in insects, which can be traced back to the neuroblast lineages that produce the Cx structure. The lineages are called type-II neuroblast (NB) lineages and are assumed to be conserved in insects. While Tribolium castaneum produces a functional larval Cx that only consists of one part of the adult Cx structure, the fan-shaped body, in *Drosophila melanogaster* a non-functional neuropile primordium is formed by neurons produced by the embryonic type-II NBs which then enter a dormant state and continue development in late larval and pupal stages.The authors present a meticulous study demonstrating that type-II neuroblast (NB) lineages are indeed present in the developing brain of Tribolium castaneum. In contrast to type-I NB lineages, type-II NBs produce additional intermediate progenitors. The authors generate a fluorescent enhancer trap line called fez/earmuff which prominently labels the mushroom bodies but also the intermediate progenitors (INPs) of the type-II NB lineages. This is convincingly demonstrated by high-resolution images that show cellular staining next to large pointed labelled cells, a marker for type-II NBs in *Drosophila melanogaster*. Using these and other markers (e.g. deadpan, asense), the authors show that the cell type composition and embryonic development of the type-II NB lineages are similar to their counterparts in *Drosophila melanogaster*. Furthermore, the expression of the *Drosophila* type-II NB lineage markers six3 and six4 in subsets of the Tribolium type-II NB lineages (anterior 1-4 and 1-6 type-II NB lineages) and the expression of the Cx marker skh in the distal part of most of the lineages provide further evidence that the identified NB lineages are equivalent to the *Drosophila* lineages that establish the central complex. However, in contrast to *Drosophila*, there are 9 instead of 8 embryonic type-II NB lineages per brain hemisphere and the lineages contain more progenitor cells compared to the *Drosophila* lineages. The authors argue that the higher number of dividing progenitor cells supports the earlier development of a functional Cx in Tribolium.While the manuscript clearly shows that type-II NB lineages similar to *Drosophila* exist in Tribolium, it does not considerably advance our understanding of the heterochronic development of the Cx in these insects. First of all, the contribution of these lineages to a functional larval Cx is not clear. For example, how do the described type-II NB lineages relate to the DM1-4 lineages that produce the columnar neurons of the Cx? What is the evidence that the embryonically produced type-II NB lineage neurons contribute to a functional larval Cx? The formation of functional circuits could rely on larval neurons (like in *Drosophila*) which would make a comparison of embryonic lineages less informative with respect to understanding the underlying variations of the developmental processes. Furthermore, the higher number of progenitors (and consequently neurons) in Tribolium could simply reflect the demand for a higher number of cells required to build the fan-shaped body compared to *Drosophila*. In addition, the larger lineages in Tribolium, including the higher number of INPs could be due to a greater number of NBs within the individual clusters, rather than a higher rate of proliferation of individual neuroblasts, as suggested. What is the evidence that there is only one NB per cluster? The presented schemes (Fig. 7/12) and description of the marker gene expression and classification of progenitor cells are inconsistent but indicate that NBs and immature INPs cannot be consistently distinguished.

We thank this reviewer for pointing out the inconsistency in our classification of cells within the lineages as one central part of our manuscript. These were due to a confusion in the used terms (young vs. immature). We have corrected this mistake and have changed the naming of the INP subtypes to immature-I and immature-II. We are confident that based on the analysed markers, type-II NBs and immature INPs can actually be distinguished with confidence.

We agree that a functional link of increased proliferation to heterochronic CX development is not shown although we consider it to be likely. As stated in the general response we have changed the manuscript to saying that the two observations (higher number of progenitors and larger lineages/more INPs) correlate but that a causal link can only be hypothesized for the time being. At the same time, we have strengthened the discussion on alternative explanations.

We would like to remain with our statement of an increased number of embryonic progeny of *Tribolium* type-II NBs. We counted the total number of progenitor cells emerging from the anterior median cluster and divided this by the number of type II NBs in that cluster. Hence, the shown increased number of cells represents an average per NB but is not influenced by the increased number of NBs. On the same line, we have never seen indication for the presence of additional NBs within any cluster while one type-II NB is what we regularly found. Hence, we are confident that we know the number of respective NBs. The fact that the fly data included also neurons and was counted at a later stage indicates that the observed differences are actually minimum estimates.

We have discussed that based on the position and comparison to the grasshopper we believe that Tribolium type-II NB 1-4 contribute to the x, y, z and w tracts. To confirm this, lineage tracing experiments would be necessary, for which tools remain to be developed.

We agree that the role of larvally born neurons and the fate of Tribolium neuroblasts through the transition from embryo to larva and pupa need to be further studied.

Available data suggests that the adult fan shaped body in Tribolium does not hugely differ in size from the *Drosophila* counterpart, although no data in terms of cell number is available. In the larva, however, no fan shaped body or protocerebral bridge can be distinguished in flies while in beetle larvae, these structures are clearly developed. Hence, we think that it is more likely that differences observed in the embryo reflect differences in the larval central complex. We discuss the need for further investigation of larval stages.

The main difference between Tribolium and *Drosophila* Cx development with regards to the larval functionality might be that *Drosophila* type-II NB lineage-derived neurons undergo quiescence at the end of embryogenesis so that the development of the Cx is halted, while a developmental arrest does not occur in Tribolium. However, this needs to be confirmed (as the authors rightly observe).

Indeed, there is evidence that cells contributing to the CX go into quiescence in flies – hence, this certainly is one of the mechanisms. However, based on our data we would suggest that in addition, the balance of embryonic versus larval proliferation of type-II lineages is different between the two insects: The increased embryonic proliferation and development leads to a functional larval CX in beetles while in flies, postembryonic proliferation may be increased in order to catch up.

**Reviewer #3 (Public Review):**
Summary:In this paper, Rethemeier et al capitalize on their previous observation that the beetle central complex develops heterochronically compared to the fly and try to identify the developmental origin of this difference. For this reason, they use a fez enhancer trap line that they generated to study the neuronal stem cells (INPs) that give rise to the central complex. Using this line and staining against *Drosophila* type-II neuroblast markers, they elegantly dissect the number of developmental progression of the beetle type II neuroblasts. They show that the NBs, INPs, and GMCs have a conserved marker progression by comparing to *Drosophila* marker genes, although the expression of some of the lineage markers (otd, six3, and six4) is slightly different. Finally, they show that the beetle type II neuroblast lineages are likely longer than the equivalent ones in *Drosophila* and argue that this might be the underlying reason for the observed heterochrony.Strengths:- A very interesting study system that compares a conserved structure that, however, develops in a heterochronic manner.- Identification of a conserved molecular signature of type-II neuroblasts between beetles and flies. At the same time, identification of transcription factors expression differences in the neuroblasts, as well as identification of an extra neuroblast.- Nice detailed experiments to describe the expression of conserved and divergent marker genes, including some lineaging looking into the co-expression of progenitor (fez) and neuronal (skh) markers.Weaknesses:- Comparing between different species is difficult as one doesn't know what the equivalent developmental stages are. How do the authors know when to compare the sizes of the lineages between *Drosophila* and Tribolium? Moreover, the fact that the authors recover more INPs and GMCs could also mean that the progenitors divide more slowly and, therefore, there is an accumulation of progenitors who have not undergone their programmed number of divisions.

We understand the difficulty of comparing stages between species, but we feel that our analysis is on the save side. At stages comparable with respect to overall embryonic development (retracting or retracted germband), the fly numbers are clearly smaller. To account for potential heterochronic shifts in NB activity, we have selected the stages to compare based on the criteria given: In *Drosophila* the number of INPs goes down after stage 16, meaning that they reach a peak at the selected stages. In Tribolium the chosen stages also reflect the phase when lineage size is larger than in all previous stages. Therefore, we believe that the conclusion that Tribolium has larger lineages and more INPs is well founded. Lineage size in Tribolium might further increase just before hatching (stage 15) but we were for technical reasons not able to look at this. As lineage size goes down in the last stage of *Drosophila* embryogenesis the number of INPs goes down and type-II NB enter quiescence, we think it is highly unlikely that the ratio between Tribolium and *Drosophila* INPs reverses at this stage, but a study of the behaviour of type-II NB in Tribolium and whether there is a stage of quiescence is still needed.

- The main conclusion that the earlier central complex development in beetles is due to the enhanced activity of the neuroblasts is very handwavy and is not the only possible conclusion from their data.

As discussed in the general response we have made several changes to the manuscript to account for this criticism and discuss alternative explanations for the observations.

- The argument for conserved patterns of gene expression between Tribolium and *Drosophila* type-II NBs, INPs, and GMCs is a bit circular, as the authors use *Drosophila* markers to identify the Tribolium cells.

We tested the hypothesis that in *Tribolium* there are type-II NBs with a molecular signature similar to flies. Our results are in line with that hypothesis. If *pointed* had not clearly marked cells with NB-morphology or *fez/erm* had not marked dividing cells adjacent to these NBs, we would have concluded that no such cells/lineages exist in the *Tribolium* embryo, or that central complex producing lineages exist but express different markers. Therefore, we regard this a valid scientific approach and hence find this argument not problematic.

An appraisal of whether the authors achieved their aims, and whether the results support their conclusions: Based on the above, I believe that the authors, despite advancing significantly, fall short of identifying the reasons for the divergent timing of central complex development between beetle and fly.

We agree that based on the available data, we cannot firmly make that link and we have changed the text accordingly.

**Recommendations for the authors:**

**Reviewer #1 (Recommendations For The Authors):**
In addition to these descriptive analyses, functional analyses can be included. RNAi is highly effective in this beetle.

We agree that functional analyses of some of the studied genes and possible effects of gene knockdowns on the studied cell lineages and on central complex development could be highly informative. However, when studying specific cell types or organs these experiments are less straight forward than it may seem as knockdowns often lead to pleiotropic effects, sterility or lethality. All the genes involved are expressed in additional cells and may have essential functions there. Given the systemic RNAi of *Tribolium*, it is challenging to unequivocally assign phenotypes to one of the cell groups. Overcoming these challenges is often possible but needs extensive optimization. Our study, though descriptive is already rich in data and is the first description of NB-II lineages in Tribolium central complex development. We see it as a basis for future studies on central complex development that will include functional experiments.

(1) IntroductionFor these reasons the beetle...Could you explain the differences in the habitats between Tribolium and *Drosophila*? or What is the biggest difference between these two species at the ecological aspect?

We have added a short characterisation of the main differences.

The insect central complex is an anterior...The author should explain why they focus on the structure.

Added

It is however not known how these temporal...If the authors want to get the answer to the question, they need to conduct functional analyses.

While we agree with the importance of functional work (see above) we believe that detailed descriptions under the inclusion of molecular markers as presented here is very informative by itself for understanding developmental processes and sets the foundation for the analysis of mutant/RNAi- phenotypes in future studies.

CX - Central complex?

We have opted to not use this abbreviation anymore for clarity.

“because intermediate cycling progenitors have also been...”Is the sentence correct?

We have included ‘INPs’ in the sentence to make clear what the comparison refers to and added a comma

“However, molecular characterization of such lineage in another...”The authors should explain why molecular characterization is necessary.

We have done so

(2) Resultsa) Figure 8. Could you delineate the skh/eGFP expression region?

We have added brackets to figure 1 panel A to indicate the extent of skh and other gene expressions within the lineages.

b) This section should be reorganized for better logical flow.

There certainly are different ways to organize this part and we have considered different structures of the results part. We eventually subjectively concluded that the chosen one is the best fit for our data (also see comment below on dpn-expression).

c) For the tables. The authors should mention what statistical analysis they have conducted.

The tables themselves are just listing the raw numbers. They are the basis for the graph in figure 9. Statistical tests (t-test) are mentioned in the legend of that figure and now also in the Methods sections.

“We also found that the large Tc-pnt...”The authors could examine the mitotic index using an anti-pH3 antibody.

We have used the anti-pH3 antibody to detect mitoses (figure 3C, table 1 and 3) but as data on mitoses based on this antibody is only a snapshot it would require a lot of image data to reliably determine an index in this specific cells. While mitotic activity over time possibly combined with live imaging might be very interesting in this system also with regards to the timing of development, for this basic study we are satisfied with the statement that the type-II NB are indeed dividing at these stages.

“Based on their position by the end of embryogenesis...”How can the authors conclude that they are neuroblasts without examining the expression of NB markers?

Type-II NB do not express asense as the key marker for type I neuroblasts. To corroborate our argument that the cells are neuroblasts we have used several criteria:

- We have used the same markers that are used in *Drosophila* to label type-II NBs (pnt, dpn, six4). We are not aware of any other marker that would be more specific.

- We have shown that these cells are larger and have larger nuclei than neighbouring cells and they are dividing

- We have shown that these cells through their INP lineages give rise to central complex neuropile

We believe that these features taken together leave little doubt that the described cells are indeed neuroblasts.

“We found that the cells they had assigned as...”How did the authors distinguish that they are really neuroblasts?

We see the difficulty that we first describe the position and development of these cells (e.g. fig 3) and then add further evidence (cell size, additional marker dpn) that these are neuroblasts (also see above). However, without previous knowledge on position (and on pnt expression as the most specific marker) the type-II NB could not have been distinguished from other NBs based on cell size or expression of other markers.

“Conserved patterns of gene expression...”This must be the first (especially dpn).

Dpn is not specific to type-II NB because it is also expressed in type-1 NBs, mature INPs and possibly other neural cells. It is therefore impossible to identify type-II NBs based on this gene alone. We therefore first used the most specific marker, pnt, in addition to adjacent fez expression to identify candidates for type-II lineages. Then we mapped expression of further genes on these lineages to support the interpretation (and show homology to the *Drosophila* lineages). Although of course the structure of a paper does not necessarily have to reflect the sequence in which experiments were done we would find putting dpn expression first misleading as it would not be clear why exactly a certain part of the expression should belong to type-II NB. Also, our pnt-fez expression data shows the position of the NB-II in the context of the whole head lobe whereas the other gene expressions are higher magnifications focussing on details. We therefore believe that the structure we chose best fits our data and the other reviewers seemed to find it acceptable as well.

“As type-II NBs contribute to central...”Before the sentence, the author could explain differences in the central complex structure between Tribolium and *Drosophila* in terms of cell number and tissue size.

We have added references on the comparisons of tissue sizes, but unfortunately there is no Tribolium data that can be directly compared to available *Drosophila* resources in terms of cell number.

“We conclude that the embryonic development of...”How did the authors conclude? They must explain their logic.Actually, before this sentence, I only found the description of the comparison between Tribolium NBs and *Drosophila* once.

We agree that this conclusion is not fully evident from the presented data. We have therefore changed this part to stating that there is a correlation with the earlier central complex development described in Tribolium. See also response to the general reviewer comments.

“Hence, we wondered...”The authors need to do a functional assessment of the genes they mentioned.

We agree that the goals originally stated at the beginning of this paragraph can only be achieved with functional experiments. We have therefore rephrased this part.

(3) Discussion“A beetle enhancer trap line...”This part should be moved elsewhere (it does not seem to be a discussion)

In accordance with this comment and reviewer#2’s similar comment we have removed this section. We have added a statement on the importance of testing the expression of an enhancer trap line to the results part and an added the use of CRISPR-Cas9 for line generation to the introduction.

“We have identified a total...”The authors emphasized that they discovered 9 type II NBs. The authors should clarify how important this it

We have added some discussion on the importance of this finding.

Dpn is a neural marker - Is this correct?

According to Bier et al 1992 (now added as reference) dpn is a pan-neural marker. Reviewer#2 also recommended calling dpn a neural marker.

“Previous work described a heterochronic...” - reference?

Reference have been added

“By contrast, we show that Tribolium...”What about the number of neurons in the central complex in Tribolium and *Drosophila*?Does the lineage size of type II NBs reflect the number?

Unfortunately, we do not have numbers for that.

**Reviewer #2 (Recommendations For The Authors):**
I recommend using page and line numbers to make reviewing and revising less timeconsuming.

We apologize for this oversight. We include a line numbering system into our resubmission.

(1) Abstract"These neural stem cells are believed to be conserved among insects, but their molecular characteristics and their role in brain development in other insect neurogenetics models, such as the beetle Tribolium castaneum have so far not been studied."I recommend explaining the importance of studying Tribolium with regard to the evolution of brain centres rather than just stating that data are lacking.

We have now emphasized the importance of Tribolium as model for the evolution of brain centres.

"Intriguingly, we found 9 type-II neuroblast lineages in the Tribolium embryo while *Drosophila* produces only 8 per brain hemisphere."It should be made clear that the 9 lineages also refer to brain hemispheres.

We have added this information

(2) IntroductionI would remove the first paragraph of the introduction; the use of Tribolium as model representative for insects is too general. The authors should focus on the specific question, i.e. the introduction should start with paragraph 2.

While we can relate to the preference for short and concise writing, we feel that giving some background on Tribolium might be important as we expect that many of our readers might be primarily *Drosophila* researchers. Keeping this paragraph also seems in line with a recommendation of reviewer#1 to add some additional information on Tribolium ecology.

"Several NBs of the anterior-most part of the neuroectoderm contribute to the CX and compared…”The abbreviation has not been introduced.

For clarity we have now opted to not use this abbreviation but to always spell out central complex.

"Several NBs of the anterior-most part of the neuroectoderm contribute to the CX and compared to the ventral ganglia produced by the trunk segments, it is of distinctively greater complexity..."Puzzling statement. Why would you compare a brain center with ventral ganglia? I recommend removing this.

We have changed this statement to just emphasizing the complexity of the brain structure.

"The dramatically increased number of neural cells that are produced by individual type-II lineages, and the fact that one lineage can produce different types of neurons..." In my opinion, this statement is too vague and unprofessional in style. Instead of "dramatically increased" use numbers.

We have removed ‘dramatically increased’ and now give a numeric example.

"The dramatically increased number of neural cells that are produced by individual type-II lineages, and the fact that one lineage can produce different types of neurons, leads to the generation of increased neural complexity within the anterior insect brain when compared to the ventral nerve cord.."I assume that this statement relates to the comparison of type I and II nb lineages. However, type I NB lineages also produce different types of neurons due to GMC temporal identity, and neuronal hemi-lineage identity.

We have rephrased and tried to make clear that the second part of the statement is not specific to type-II NB only. In line with the comment above we have also removed the reference to the ventral nerve cord.

"In addition, in *Drosophila* brain tumours have been induced from type-II NBs lineages [34], opening up the possibility of modelling tumorigenesis in an invertebrate brain, thus making these lineages one of the most intriguing stem cell models in invertebrates [35,36]."This statement is misplaced here; it should be mentioned at the start (if at all).

We have moved this statement up.

"However, molecular characterisation of such lineages in another insect but the fly and a thorough comparison of type-II NBs lineages and their sub-cell-types between fly and beetle are still lacking"The background information should include what is known about type-II NB lineages in Tribolium, including marker gene expression, e.g. Farnworth et al.

We refer to He et al 2019, Farnworth et al 2020 and Garcia-Perez 2021. All these publications speculate about a contribution of type-II NBs to Tribolium central complex development but do not show evidence of it. As we emphasize throughout the manuscript, the present work is the first description of type-II NB in Tribolium.

"The ETS-transcription factor pointed (pnt) marks type-II NBs [40,41], which do not express the type-I NB marker asense (ase) but the pro-neural gene deadpan (dpn)" Deadpan is considered a pan-neural gene. To avoid confusion, I would remove "proneural" throughout.

We have done so throughout the manuscript.

"We further found that, like the type-II NBs itself, the youngest Tc-pnt-positive but fezmm-eGFP-negative INPs neither express Tc-ase (Fig. 5D, pink arrowheads)." What is the evidence that these are the youngest pnt positive cells? Position? This needs to be explained.

We have clarified that ‘youngest pnt-positive cells’ refers to the position of these cells close to the type-II NB.

"Therefore these neural markers can be used for a classification of type II NBs (Tc-pnt+, Tcase-), young INPs (Tc-pnt+, Tc-fez/erm-, Tc-ase-), immature INPs (Tc-pnt+, Tcfez/erm+, Tcase+), mature INPs (Tc-dpn+, Tc-ase+, Tc-fez/erm+, Tc-pros+), and GMCs (Tc-ase+, Tcfez/ erm+, Tc-pros+, Tc-dpn). This classification is summarized in Fig. 7 A-B."This is not the best classification and not in line with the schemes in Figure 7 - the young INPs are also immature. What is the difference? It needs to be explained what "mature" means (dividing?).

Thank you for pointing this out. We have corrected the error in this part that confused the two original groups (young and immature). To take the immaturity of both types of INPs into account we have then also changed our naming of INP subtypes into immature-I and immature-II and throughout the manuscript. Figure 7 and figure 12 were also changed accordingly. While our classification if primarily based on gene expression the available data indicates that both types of immature INPs are not dividing, whereas mature INPs are. We have added a statement on that to this part.

"In beetles a single-unit functional central complex develops during embryogenesis while in flies the structure is postembryonic."This statement is vague - the authors need to explain what is meant by "single-unit". The phrase "The structure is postembryonic" also needs more explanation. The *Drosophila* CX neuroblasts lineages originate in the embryo and the neurons form a commissural tract that becomes incorporated into the fan-shaped body of the Cx.

We have explained single-unit central complex and have improved our summary of known differences in central complex development between fly and beetle.

"To assess the size of the embryonic type-II NBs lineages in beetles we counted the Tc- fez/erm positive (fez-mm-eGFP) cells (INPs and GMCs) associated with a Tc-pntexpressing type-II NBs of the anterior medial group (type-II NBs lineages 1-7). It is not clear what is meant by "with a Tc-pnt-expressing type-II NBs". Is this a typo?"

We have removed this bit.

(3) DiscussionI would remove the first paragraph "A beetle enhancer trap lines reflects Tc-fez/earmuff expression". This is a repetition of the methods rather than a discussion.

This part has been removed also in line with reviewer#1’s comment.

(4) FiguresFigure 2To which developing structure do the strongly labelled areas in Figure 2D correspond?

We believe that these areas from the protocerebrum including central complex, mushroom bodies and optic lobe. We have added this to the text and to the figure legend.

Figure 7What do A and B represent? Different stages?

A and B show the same lineage but map the expression of different additional markers for clarity. We have added an explanation of this.

The classification contradicts the description in the section "Conserved patterns of gene expression mark Tribolium type-II NBs, different stages of INPs and GMCs" (last sentence) where young INPs are first in the sequence and described as pnt+, erm-, ase- and immature INPs as pnt+ erm+ and ase+.

We have corrected this mistake and changed the names of the subtypes into immatureI and immature-II (see above).

"We conclude that the evolutionary ancient six3 territory gives rise to the neuropile of the z, y, x and w tracts."Please clarify if six3 is also expressed in the corresponding grasshopper NB lineages or if your conclusion is based on the comparison of *Drosophila* and Tribolium and you assume that this is the ancestral condition.

Six3 expression has not been studied in grasshoppers. Owing to the highly conserved nature of an anterior median six3 domain in arthropods and bilaterian animals in general, we would expect it to be expressed anterior-medially in grasshoppers as well. In *Drosophila* the gene is expressed in the anterior-medial embryonic region where the type-II NBs are expected to develop, but to our knowledge it has not been specifically studied which type-II NB lineages are located within this domain. We have clarified in our text that we do not claim that the origin of anterior-medial type-II NB 1-4 and the X,Y, Z and W lineages from the six3 territory is highly conserved but only the territory itself. As far as we know our work is the first to analyse the relationship of type-II lineages and the conserved head patterning genes *six3* and *otd*. We have added some clarification of this into this part of the discussion.

(5) MethodsThe methods section should include the methods for cell counting, as well as cell and nuclei size measurements including statistics (e.g. how many embryos, how many NB lineages). The comparison of the Tribolium NB lineage cell numbers to published *Drosophila* data should include a brief description of the method used in *Drosophila* (in addition to the method used here in Tribolium) so that the reader can understand how the data compare.

We have added a separate section on this to the Methods part which also includes the criteria used in *Drosophila*. We have also included some more information to the results part on the inclusion of neurons in the *Drosophila* counts that may only be partially included in our numbers. This does however not change the results in terms of larger numbers of progenitor cells in *Tribolium*.

(6) Typos and minor errorsAbstract“However, little is known on the developmental processes that create this diversity”Change to ... little is known about

Changed.

NBs lineagesChange to NB lineages throughout.

We have used text search to find and replace all position where this was used erroneously,

Results"Schematic drawing of expression different markers in type-II NB lineages.."Schematic drawing of expression of different markers

Corrected

Discussion"However, the type-II NB 7, which is we assigned to the anterior medial group but which...".... which we assigned....

corrected

"......might be the one that does not have a homologue in the fly embryo The identification of more..." Full stop missing.

Added.

"Adult like x, y, and w tracts as well as protocerebral bridge are...."Change to "The adult like x, y, and w tracts as well as the protocerebral bridge are....

This part has been removed with the rewriting of this paragraph.

**Reviewer #3 (Recommendations For The Authors):**
(1) Suggestions for improved or additional experiments, data, or analyses:a) The analysis of nuclear size is wrong. The authors compare the largest cell of a cluster of cells with a number of random cells from the same brain. It is obvious that the largest cell of a cluster will be larger than the average cell of the same brain. A better control would be to compare the largest cell of the pnt+ cluster with the largest cell of a random sample of cells, although this also comes with biases. Personally, I have no doubt that the authors are looking at neuroblasts, based on the markers they are using, so I would recommend completely eliminating Figure 4.

We agree that we produced a somewhat biased and expected result when we select the largest cell of a cluster for size comparison. However, we found it important to show based on a larger sample that these cells are also statistically larger than the average cell of a brain, which we think our assessment shows. We do not claim that type-II NBs are the largest cells of a brain, or that they are larger than type-I NBs, therefore in a random sample there might be cells that are equally big (see also distribution of the control sample shown in figure 4, and we have added a note on this to the text). We are happy to hear that this reviewer has no doubts we are looking at neural stem cells. However, reviewer#1 did express some hesitations and therefore we think it is important to keep the information on cell size as part of our argument that we are indeed looking at type-II NBs (gene expression, cell size, dividing, part of a neural lineage).

b) The comparison of NB, INP, and GMC numbers between *Drosophila* and Trbolium (section "The Tribolium embryonic lineages of type-II NBs are larger and contain more mature INPs than those of *Drosophila*") compares an experiment that the authors did with published data. I would suggest that the authors repeat the *Drosophila* stainings and compare themselves to avoid cases of batch effects, inconsistent counting, etc.

None of the authors is a *Drosophila* expert or has any experience at working with this model and reassessing the lineage size would require a number of combinatorial staining. Therefore, we feel that using the published data produced by experts and which also includes repeat experiments is for us the more reliable approach.

c) In Figure 10, there are some otd+ GFP+ cells laterally. What are these?

We believe that these cells contribute to the eye anlagen. We have added this information to the legend.

(2) Minor corrections to the text and figures:a) There are some typos in the text: e.g. "pattering" in the abstract.

We have carefully checked the text for typos and hope that we have found everything.

b) The referencing of figures in the text is inconsistent (eg "Figure 5 panel A" vs "Figure 5D" on page 12).

We have checked throughout the manuscript and made sure to always refer to a panel correctly.

c) In Figure 3C, the white staining (anti-PH3) is not indicated in the Figure.

The label has been added in the figure.

d) Moreover, in Figure 3, green is not very visible in the images.

We have improved the colour intensity where possible.

e) In the figures, it might be better to outline the cells with color-coded dashed circles instead of using arrows.

We think that this would obscure some details of the stainings and create a rather artificial representation. We also feel that doing this consistently in all our images is an amount of work not justified by the degree of expected improvement to the figures

NOTE: We are submitting a revised version of the supplementary material which only contains two minor changes: a headline was added to Table S4 (Antibodies and staining reagents) and a typo was corrected in line one of table S5 (TC to Tc).